# Evaluating the potential of wake steering co-design for wind farm layout optimization through a tailored genetic algorithm

Matteo Baricchio[1], Pieter M. O. Gebraad[2], and Jan-Willem van Wingerden[1]

[1]Delft Center for System and Control, Faculty of Mechanical Engineering, Delft University of Technology, Delft, the Netherlands
[2]Youwind Renewables, Barcelona, Spain

**Correspondence:** Matteo Baricchio (m.baricchio@tudelft.nl)

**Abstract.** Wake steering represents a viable solution to mitigate the wake effect within a wind farm. New research that consider the effect of the control strategy within the layout optimization are emerging, adopting a co-design approach. This study estimates the potential of this technique within the layout optimization for a wide range of realistic conditions. To capture the benefits of such method, a genetic algorithm tailored to the layout optimization problem has been developed in this work, hence appointed as layout-optimization genetic algorithm (LO-GA). The crossover phase is designed to recognize and exploit the differences and the similarities between parent layouts whereas the randomness of the mutation is limited to improve the exploration of the design space. New relations have been introduced to calculate the geometric yaw angles based on the reciprocal positions between the turbines. For a base case of 16 turbines located at Hollandse Kust Noord site, a gain in the annual energy production (AEP) between $0.3\%$ and $0.4\%$ is obtained when the co-design approach is adopted. This increases up to $0.6\%$ for larger farms, saturating after 25 turbines. However, the benefit of the co-design decreases in case of low power densities or if the wind resource is highly unidirectional. On the other hand, in case wake steering is not applied during the operation of the farm, a decrease in the AEP up to $0.6\%$ is registered for a layout optimized with the co-design method. To prevent the risk related to future decisions on the control strategy, a multi-objective co-design approach is proposed. This is based on the simultaneous optimization of the layout for a scenario in which wake steering is applied versus a case where wake steering is not adopted during the operation of the farm. We have concluded that the solutions obtained with this method ensure an AEP gain higher than $0.3\%$ for a 16-turbines farm while limiting the loss below $0.1\%$ in case wake steering is not applied. However, these AEP gains are affected by the size of the wind direction bins adopted in the simulations, enhancing the necessity of taking into account the wind direction errors and the yaw actuation constraints for a realistic evaluation of the co-design approach.

## 1 Introduction

The mitigation of the wake interaction between wind turbines represents one of the major challenges within the design and the operation of wind farms (Meyers et al., 2022). Higher power generation and load reduction can be achieved by minimizing the wake effect, increasing the revenues associated to the electricity production while extending the lifetime of the farm (Cassamo, 2022). The rapid development of offshore wind energy has urged the necessity of constraining a large number of turbines in

limited areas, increasing the impact of these effects (Pettersen et al., 2023). Therefore, innovative solutions are required to address the wake interactions in order to extract the topmost value from a wind farm.

The minimization of the wake losses is generally addressed by selecting appropriately the positions of the turbines within the available surface (Mosetti et al., 1994). This design phase is usually referred to as wind farm layout optimization problem (WFLOP) and aims to maximize one or multiple objectives while satisfying various types of constraints, e.g. geographical restrictions or minimum spacing (Feng and Shen, 2015). Different implementations of WFLOP can be distinguished depending on how the positions of the turbines are related to the optimization variables. Specifically, they can be parameterized through a limited number of variables by introducing regular layouts, where periodic patterns are repeated throughout the farm. Otherwise, the positions of the turbines can be identified by discrete or continuous coordinates, depending on the requirements on the resolution of the design space. Therefore, WFLOP can assume different natures, whose choice often depends on the trade-off between the required accuracy and the computational cost, usually determined by the size of the farm and the level of fidelity of the models adopted to calculate the wake interactions. Depending on the purpose of the study, different objectives can be considered for the WFLOP. However, current methods mainly focus on the maximization of the annual energy production (AEP) or the minimization of the levelized cost of energy (LCOE) (Tao et al., 2020).

Wind farm control represents another viable solution to mitigate the wake effects during the operation of the plant, based on the performance optimization of the entire farm considered as a one entity instead of a summation of individual optimized turbines (van Wingerden et al., 2020). Relying on various concepts, different wind farm control techniques have been developed in the recent years (Meyers et al., 2022). Among these approaches, wake steering has been demonstrated to improve significantly the power production of a wind farm, deviating the wakes from the downstream turbines by actuating yaw control (Doekemeijer et al., 2021).

The design phase of a wind farm is often not influenced by the wind farm control technique, which are only considered during the operation of the plant (Stanley et al., 2023). However, different studies have proved that taking into account the wind farm control strategy already during the design stage could lead to significant improvements in the performance, especially within the WFLOP. This is usually referred to as the co-design approach, in contrast to the traditional sequential method in which design and operation phases constitute two separate blocks that are optimized individually (Fleming et al., 2016).

The co-design approach is included in the WFLOP by adapting the control variables during the computation of the objective function, e.g. the axial induction factor for static induction control or yaw angles for wake steering (Stanley et al., 2023). Aiming to optimize the wind farm layout considering its entire lifetime, the optimal control variables have to be determined for each possible case experienced by the farm, i.e. different combinations of wind speeds and directions. However, this leads to an optimization problem characterized by an extremely high number of variables, requiring expensive computational resources in terms of core numbers and/or simulation time (Fleming et al., 2016).

Various solutions have been introduced to tackle the "curse of dimensionality" while capturing the benefits of the co-design approach. Fleming et al. (2016) and Yin et al. (2023a) have decoupled the optimization problem, i.e. the control variables are optimized on an initial layout and then these values are assumed within the WFLOP. A nested approach has been proposed by Pedersen and Larsen (2020), who have iterated for each step the calculation of the optimal coordinates followed by the

optimization of the control variables. Another nested method has been adopted by Chen et al. (2022), decomposing the WFLOP for different wind scenarios and constraining the same turbines' positions through a coordination problem. Another possible strategy consists in the use of regular layouts to reduce the number of variables related to the WFLOP without affecting those related to the control optimization (Hou et al., 2017). This approach can be followed by a position refinement, as implemented by Tang et al. (2022). Alternatively, the number of variables can be diminished by limiting the simulations to one representative wind speed value for each wind direction (Gebraad et al., 2017; Song et al., 2023). Machine learning surrogate models are also implemented to fasten the computation of the objective function or to improve the exploration of the highly-dimensional design space (Song et al., 2023; Yin et al., 2023b).

Recently, a novel approach has been introduced by Stanley et al. (2023) where analytical relations are used to obtain the optimal control variables within the AEP calculation, avoiding an expensive nested optimization. Specifically, this study focuses on the wake steering technique and the yaw angle of each turbine is determined for each flow case based on the position of the downstream turbines, hence this approach is referred to as geometric yaw. This approximation of the optimal yaw angles has also been implemented in the open-source software FLORIS (National Renewable Energy Laboratory, 2024). In the study of Stanley et al. (2023), this method enables the integration of the wake steering within the WFLOP, increasing the AEP up to $0.8\%$ with respect to the traditional sequential approach. However, this value refers to a rather specific farm consisting of 16 wind turbines simulated in a site characterized by a Gaussian hill spatially varying inflow. Moreover, high power density and low minimum distance spacing between turbines have been adopted. As mentioned by Stanley et al. (2023), these conditions boost the benefits of the co-design approach, coming close to the maximum improvement achievable with the proposed method. On the other hand, the geometric yaw relation introduced by Stanley et al. (2023) is based on a limited number of variables, enabling a straightforward interpretation and implementation. Therefore, there is a significant margin of improvement that could enhance the advantage of the co-design approach for wake steering. These considerations demonstrate the necessity to understand the real potential of the geometric yaw method within the WFLOP for more realistic conditions.

A crucial aspect of the WFLOP is the choice of the optimization algorithm to extract the optimal positions of the turbines. The literature lacks a full agreement on the most appropriate optimization algorithm and both gradient-based (GB) and gradient-free (GF) techniques are adopted to solve the WFLOP (Thomas et al., 2023). The non-convexity of WFLOP challenges conventional GB methods in reaching the global optimum, requiring multiple runs from a variety of starting conditions (Guirguis et al., 2016). An attempt to reduce the multi-modality of the problem has been made by Thomas et al. (2022), who introduced a technique named wake expansion continuation, based on the gradual reduction of the wake diameter during each iteration. Moreover, GB methods cannot guarantee high performance on black-box objective functions, which would require the computationally expensive finite differences for the calculation of the gradient (Martins and Ning, 2022). On the other hand, GF approaches such as genetic algorithm (GA) and particle swarm optimization (PSO) are usually favorable in case of a design space characterized by many local optima. However, these methods could lead to a higher number of function calls than GB and tend not to scale effectively with high number of variables (Rios and Sahinidis, 2013). Sparse nonlinear optimizer (SNOPT) (Gill et al., 2005) is often adopted for WFLOP and it has been used by Stanley et al. (2023) to test the co-design approach using the geometric yaw relation. However, the study from Thomas et al. (2023) has compared eight

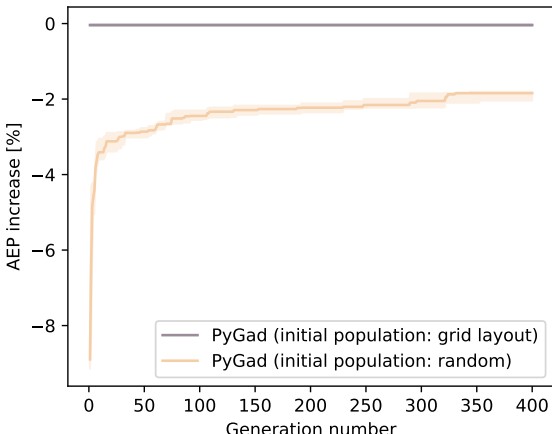

**Figure 1.** WFLOP using PyGad GA: comparison between the use of a random initial population and a population generated from a regular layout. The solid lines represent the median AEP increase with respect to the regular layout, whereas the areas refer to the range between the $25^{th}$ and $75^{th}$ percentiles, being the results of multiple simulations of the same case.

promising optimization algorithms, including SNOPT, and has concluded that the best performance are achieved by a discrete exploration-based optimization (DEBO) method, which combines a greedy initialization and discrete refinement of the solution, developed specifically for the WFLOP.

The popularity of GA to solve the WFLOP is due to its ability to explore the design space with a high degree of solutions' variety. However, as the number of turbines increases, the capability of convergence is seriously affected. This is demonstrated
in Fig. 1, which shows the results of a basic implementation of the GA using the open-source python library PyGad (Gad, 2023). In this figure, the case where a regular layout is used to create an initial population is compared to a random initialization of the optimization variables. It can be observed that in the former case the GA is not able to further improve the initial layout whereas in the latter the GA cannot converge to a solution better than a regular layout. This behavior is related to the excessive randomness that characterizes the exploration of the design space. Therefore, this study aims to exploit the ability of the
GA to explore a non-convex design space while improving its convergence ability by capturing the physical meaning of the optimization variables. This is achieved by developing novel method named layout-optimization genetic algorithm (LO-GA), where the selection, crossover and mutation phases are designed specifically for the WFLOP.

The contribution of this work is fourfold:

- New geometric yaw relations are formulated to improve the capability of approximating the optimal yaw angles.

- The effectiveness of the co-design approach is tested for different power densities, farm sizes and site types, to understand the potential of this method in realistic conditions.

- The impact of not applying wake steering for a layout optimized using the co-design method is quantified and a multi-objective co-design approach is investigated.

- A tailored genetic algorithm for the WFLOP is developed, referred to as LO-GA.

The remainder of the paper is structured as follows. In sections 2-5 the methodology adopted in this work is explained, describing the LO-GA and the geometric yaw relations developed in this study, as well as introducing the case studies that have been selected. Then, section 6 includes the results that quantify the potential of the co-design approach. These results are then discussed in section 7 whereas section 8 draws the conclusions and includes the recommendations for future work.

## 2  Co-design within wind farm layout optimization

This section explains how the wind farm layout optimization can be solved adopting a co-design approach. First, the WFLOP is defined specifying objectives and constraints considered in this study. Second, the methodology adopted to apply and evaluate the co-design concept within the WFLOP is described.

### 2.1  Wind farm layout optimization problem

WFLOP consists in optimizing the position of the turbines within a pre-defined area. In this study, the objective of the WFLOP is the maximization of the AEP, calculated as shown in Eq. 1 by summing the power ($P_{\theta,u}$) generated by the farm for every wind direction ($\theta$) and speed ($u$) multiplied by the correspondent probability of occurrence ($\rho_{\theta,u}$). This is referred as the objective or fitness function. The optimization variables are identified by the Cartesian coordinates of the turbines ($x, y$), hence the total number of variables is equal to $2\,n_{wt}$, with $n_{wt}$ indicating the total number of turbines. The turbines' positions are restricted to a rectangular area, as expressed in Eq. 2. Moreover, a spacing constraint is considered to guarantee a minimum distance ($d_{\min}$) between the turbines, formulated in Eq. 3.

$$\max_{\mathbf{x},\mathbf{y}} AEP(\mathbf{x},\mathbf{y}) = \max_{\mathbf{x},\mathbf{y}} 8760\,\mathrm{h\,yr}^{-1} \cdot \sum_{\theta=0}^{N_\theta} \sum_{u=0}^{N_u} \rho_{\theta,u} \cdot P_{\theta,u}(\mathbf{x},\mathbf{y}) \tag{1}$$

$$x_i \in [x_{\min}, x_{\max}],\ y_i \in [y_{\min}, y_{\max}]\ \forall i,j \in n_{wt} \tag{2}$$

$$\sqrt{(x_i - x_j)^2 + (y_i - y_j)^2} \geq d_{\min}\ \forall i,j \in n_{wt}\ \text{s.t.}\ i \neq j \tag{3}$$

The AEP of the wind farm is computed using PyWake (Pedersen et al., 2023), an open-source tool developed by the Technical University of Denmark (DTU) which simulates the wake interaction between the turbines of a wind farm. The Bastankhah Gaussian Deficit (Bastankhah and Porté-Agel, 2014) model is selected in this study to calculate the wake deficit whereas the wake deflection is calculated according to Jiménez et al. (2010).

**Table 1.** Hyperparameters of LO-GA

| Hyperparameter name | Symbol |
|---|---|
| Number of generations | $n_{\text{gen}}$ |
| Size of the population | $n_{\text{pop}}$ |
| Number of parent solutions to keep | $n_{\text{keep}}$ |
| Percentage of selection | $p_s$ |
| Percentage of mutation | $p_m$ |
| Step of mutation | $s_m$ |
| Distance limit | $d_{\text{lim}}$ |

## 2.2 Co-design approach

Within the WFLOP, the co-design approach consists in considering the control strategy of the wind farm while computing the objective function. In this case, the wind farm control strategy is limited to the wake steering whereas the objective consists in the AEP calculation. Yaw angles are therefore specified for each wind speed and direction bin while computing the AEP through the PyWake function.

In this study, the improvement obtained through the co-design approach is determined as follows. First, the wind farm layout optimization is performed for both cases, namely neglecting and considering wake steering through the geometric yaw relations, obtaining two different layouts. The same starting layout is adopted to avoid the influence of different initial conditions. Second, an accurate yaw optimization is computed for both layouts, determining the optimal angles for each wind speed and direction. Specifically, the serial-refine yaw optimization method is adopted in this study for this phase (Fleming et al., 2022). Then, the AEP is calculated for both layouts considering the optimal yaw angles obtained in this last step. These AEP values are finally compared, expecting the wind farm layout obtained through the co-design approach to outperform the layout resulted from the traditional method.

## 3 Layout optimization genetic algorithm (LO-GA)

A genetic algorithm named LO-GA tailored to the WFLOP is developed in this work, where new methods are introduced specifically for this optimization problem. These are explained in the next sections following the main blocks that constitute the classic implementation of a GA, namely initialization of the population, selection, crossover and mutation. Table 1 includes all the hyperparameters required by the algorithm, which will be described in the next sections in detail along with their tuning phase. Specifically, some of these hyperparameters allow different values depending on the generation number, enabling dynamic selection and dynamic mutation (Hassanat et al., 2019). An overview of LO-GA is included in Fig. 2, where the main blocks, i.e. selection, crossover and mutation, are highlighted.

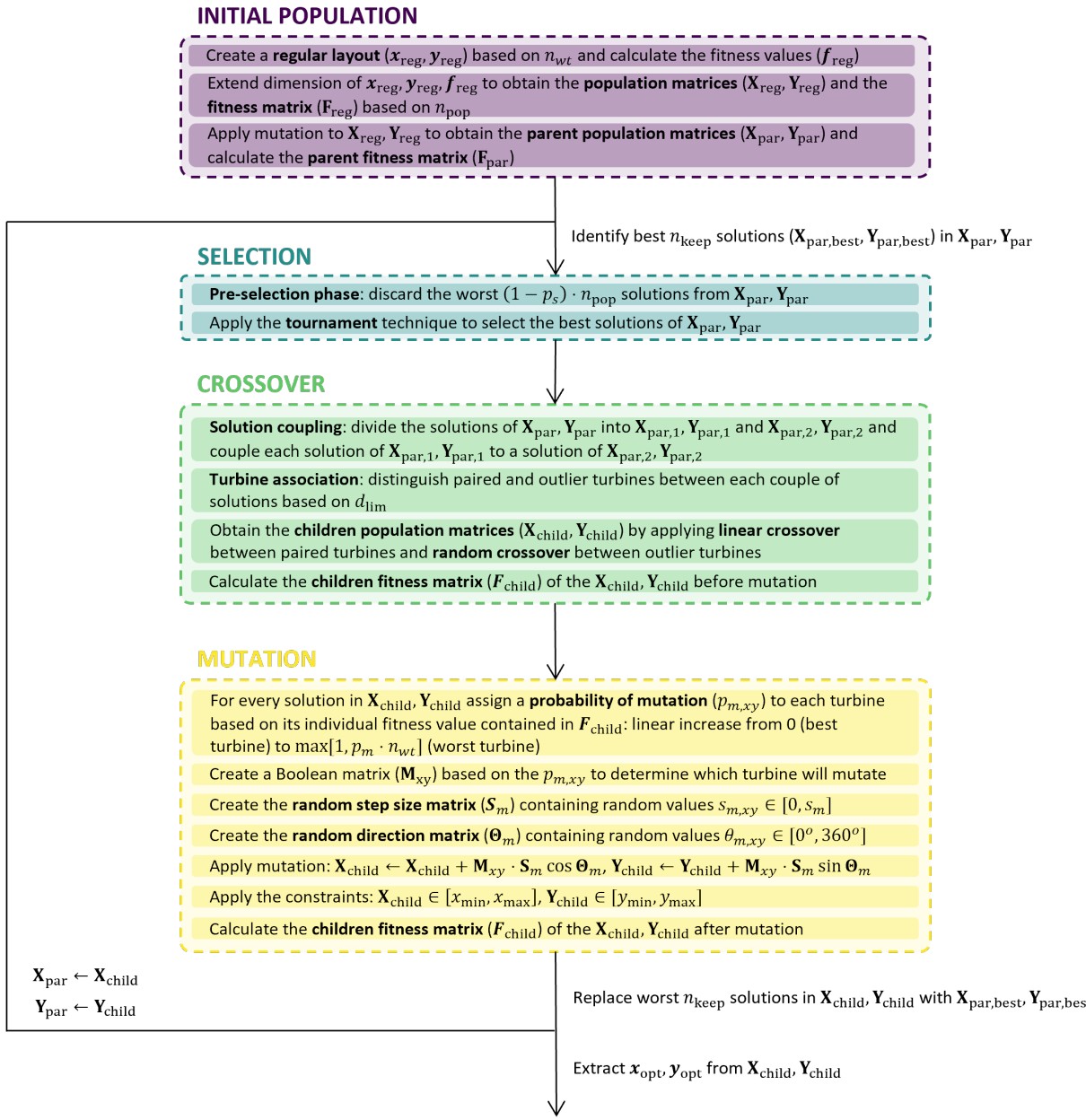

**Figure 2.** Layout optimization genetic algorithm (LO-GA)

## 3.1 Initial population

The starting point of a GA consists in providing an initial population of solutions that enables the algorithm to converge towards an optimal solution. Therefore, the initial population has to be sufficiently close to the optimal solution while preserving

the randomness required to further improve such starting solution. Specifically, an initial population with a low degree of randomness would achieve only limited improvements whereas an excessive degree of randomness would require an unfeasible number of the generations for the algorithm to converge. In this study, a regular layout $(\boldsymbol{x}_{\mathrm{reg}}, \boldsymbol{y}_{\mathrm{reg}})$ is generated based on the

165 number of turbines $(n_{wt})$, where these occupy the positions of a squared grid. Such layout is duplicated $n_{wt}$ times in order to form the population matrices $(\mathbf{X}_{\mathrm{reg}}, \mathbf{Y}_{\mathrm{reg}})$. These matrices will be the base unit for the LO-GA iterations, where different solutions are contained along the first dimension (axis 0) whereas the coordinates of each solution are present in the second dimension (axis 1). To include the randomness within the initial population a mutation step is applied, obtaining the parent population matrices $(\mathbf{X}_{\mathrm{par}}, \mathbf{Y}_{\mathrm{par}})$. The mutation phase is described in detail in section 3.4, where it is mentioned that it requires

the fitness value of the solutions, i.e. the AEP in this study. Specifically, the fitness value of each individual turbine is needed. Such calculation is enabled by the PyWake method for AEP calculation and it is performed for $\mathbf{X}_{\mathrm{reg}}, \mathbf{Y}_{\mathrm{reg}}$. These values are stored in the fitness matrix $(\mathbf{F}_{\mathrm{reg}})$, which has the same structure of $\mathbf{X}_{\mathrm{reg}}, \mathbf{Y}_{\mathrm{reg}}$. Lastly, the fitness values are computed for the parent population matrices $(\mathbf{X}_{\mathrm{par}}, \mathbf{Y}_{\mathrm{par}})$ and stored in the parent fitness matrix $(\mathbf{F}_{\mathrm{par}})$. In conclusion, $\mathbf{X}_{\mathrm{par}}, \mathbf{Y}_{\mathrm{par}}$ and $\mathbf{F}_{\mathrm{par}}$ represent the starting point for the LO-GA iterations described in the following sections.

## 3.2 Selection

The selection phase developed in the LO-GA is divided into two different steps. First, a pre-selection is applied, where the $(1 - p_s) \cdot n_{\mathrm{pop}}$ solutions characterized by the lowest fitness are discarded. The fitness of the solutions is expressed through the fitness parent vector $(\boldsymbol{f}_{\mathrm{par}})$, obtained by the sum of the AEP of all the turbines for each layout in the population. After this pre-selection phase, the tournament technique is adopted to extract the best solutions while ensuring a sufficient degree of

180 randomness (Miller and Goldberg, 1995). Therefore, the outputs of this phase are new parent population matrices $(\mathbf{X}_{\mathrm{par}}, \mathbf{Y}_{\mathrm{par}})$ with the selected solution, along with their correspondent fitness parent matrix $(\mathbf{F}_{\mathrm{par}})$.

## 3.3 Crossover

The purpose of the crossover phase is to generate children solutions from the parent population, aiming to capture and combine the optimal characteristics of each parent. This represents the main difference with respect to the traditional genetic algorithms

where the crossover is determined by the random combination of the parent solutions. On the other hand, the crossover phase integrated in the LO-GA relies on the fitness value of every individual turbine to prevent an excessive randomness during the design space exploration. The first step consists in dividing the parent population into two different parent matrices, i.e. obtaining $\mathbf{X}_{\mathrm{par},1}, \mathbf{Y}_{\mathrm{par},1}$ and $\mathbf{X}_{\mathrm{par},2}, \mathbf{Y}_{\mathrm{par},2}$ starting from $\mathbf{X}_{\mathrm{par}}, \mathbf{Y}_{\mathrm{par}}$. Meanwhile, the values of the parent fitness matrices $\mathbf{F}_{\mathrm{par},1}, \mathbf{F}_{\mathrm{par},2}$ are inherited from $\mathbf{F}_{\mathrm{par}}$ since the parent solutions have not been modified yet. Then, each solution contained in

$\mathbf{X}_{\mathrm{par},1}, \mathbf{Y}_{\mathrm{par},1}$ is coupled with a solution of $\mathbf{X}_{\mathrm{par},2}, \mathbf{Y}_{\mathrm{par},2}$ and the actual crossover phase starts. This is applied to each couple of solutions and is divided into two different steps, namely the turbine association and the linear/random crossover, where for every couple of parent solutions two different children solutions are generated. First, the turbine association aims to understand the similarities of the two coupled layouts, labelling each turbine as paired or outliers. In case a turbine from the first layout is positioned within a distance lower than $d_{\mathrm{lim}}$ from a turbine of the second layout, these two turbines are labelled as paired. The

condition that a turbine of the first layout can be paired with a maximum of one turbine of the second layout, and vice versa, is enforced. Otherwise, the turbines are labelled as outliers. An example of this process of turbine association is depicted in Fig. 3.

The second step consists in applying two novel techniques developed specifically for this optimization problem: 'linear crossover' between paired turbines and 'random crossover' between the outliers. The former technique aims to combine the positions of the two paired turbines, identified by $(x_{p,1}, y_{p,1})$ and $(x_{p,2}, y_{p,2})$, in order to generate two different children turbines, identified by $(x_{c,1}, y_{c,1})$ and $(x_{c,2}, y_{c,2})$, contained in the first and in the second children layouts and positioned along the line that connects the parent turbines. Specifically, $(x_{c,1}, y_{c,1})$ is placed within the parent turbines whereas $(x_{c,2}, y_{c,2})$ on the side of the parent turbine characterized by the highest fitness value. Indicating with $f_1$ and $f_2$ the fitness values of the two paired parent turbines extracted from $\mathbf{F}_{\mathrm{par},1}, \mathbf{F}_{\mathrm{par},2}$, the coordinates of the children turbines are calculated as shown in Eq. 4-6. Therefore, the children turbines move closer to the parent turbine characterized by the higher fitness. This process is shown through an example in Fig. 3. Lastly, to complete the children population, a random crossover is applied between the outlier turbines. This means that the remaining turbines for each children layouts are selected randomly among the outlier turbines of the parent coupled layouts.

$$
\begin{cases}
x_{c,1} = \dfrac{f_1}{f_1 + f_2} \cdot x_{p,1} + \dfrac{f_2}{f_1 + f_2} \cdot x_{p,2} \\
y_{c,1} = \dfrac{f_1}{f_1 + f_2} \cdot y_{p,1} + \dfrac{f_2}{f_1 + f_2} \cdot y_{p,2}
\end{cases}
\tag{4}
$$

$$
\begin{cases}
x_{c,2} = x_{p,1} + (x_{p,1} - x_{p,2}) \cdot \dfrac{f_1}{f_1 + f_2} \\
y_{c,2} = y_{p,1} + (y_{p,1} - y_{p,2}) \cdot \dfrac{f_1}{f_1 + f_2}
\end{cases}
\quad \text{if } f_1 \geq f_2
\tag{5}
$$

$$
\begin{cases}
x_{c,2} = x_{p,2} + (x_{p,2} - x_{p,1}) \cdot \dfrac{f_2}{f_1 + f_2} \\
y_{c,2} = y_{p,2} + (y_{p,2} - y_{p,1}) \cdot \dfrac{f_2}{f_1 + f_2}
\end{cases}
\quad \text{if } f_2 > f_1
\tag{6}
$$

### 3.4 Mutation

After the crossover phase the children solutions are compacted into the children population matrices $\mathbf{X}_{\mathrm{child}}, \mathbf{Y}_{\mathrm{child}}$, which will be mutated to foster the diversity and the randomness of the solutions. Specifically, adaptive mutation is applied in this study while keeping the physical meaning of the design variables, i.e. spatial coordinates (Marsili Libelli and Alba, 2000). However, to apply an adaptive mutation the fitness values of the children solutions have to be calculated hence the fitness children matrix ($\mathbf{F}_{\mathrm{child}}$) is computed. Specifically, the concept of adaptive mutation is applied within each individual turbine of the children layouts through assigning a different probability of mutation to every turbine. Therefore, for every solution, the turbines are

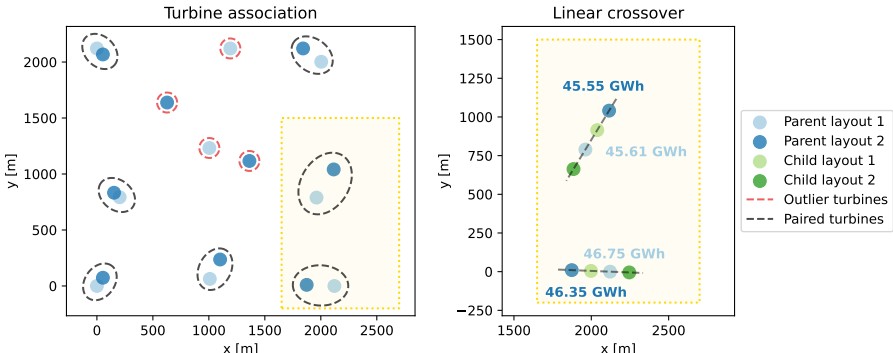

**Figure 3.** The figure on the left shows an example of the turbine association phase. The figure on right depicts the linear crossover technique for a limited number of turbines, highlighted by the yellow area. This example does not represent an optimized layout used in this study but it has the only purpose of clarifying the process adopted to apply these techniques.

sorted based on their fitness, and a value increasing linearly from $0$ (best turbine) to $\max\left[n_{wt} \cdot p_m, 1\right]$ (worst turbine) is assigned to every turbine. Based on this probability of mutation, a boolean matrix ($\mathbf{M}_{x,y}$) is created to determine which of the turbines of $\mathbf{X}_{\mathrm{child}}, \mathbf{Y}_{\mathrm{child}}$ will mutate. Simultaneously, the matrices containing the step of mutation ($\mathbf{S}_m$) and direction of mutation ($\mathbf{\Theta}_m$) are created by generating random values within $[0, s_m]$ and $[0°, 360°]$, respectively. These values differ for each turbine in the children population matrices. Then, Eq. 7 is applied to perform the mutation of the children population. The presence of $\mathbf{M}_{x,y}$ ensures that only a limited number of turbines will mutate irrespective of the values of $\mathbf{S}_m$ and $\mathbf{\Theta}_m$. This type of mutation limits the degree of randomness but ensures that the new mutated solutions will not differ significantly from the optimal layout, obtaining a faster convergence. An example of this mutation process is depicted in Fig. 4.

$$
\begin{cases}
\mathbf{X}_{\mathrm{child}} = \mathbf{X}_{\mathrm{child}} + \mathbf{M}_{x,y} \cdot \mathbf{S}_m \cdot \cos(\mathbf{\Theta}_m) \\
\mathbf{Y}_{\mathrm{child}} = \mathbf{Y}_{\mathrm{child}} + \mathbf{M}_{x,y} \cdot \mathbf{S}_m \cdot \sin(\mathbf{\Theta}_m)
\end{cases}
\tag{7}
$$

The last step of the LO-GA is to compute the fitness calculation of the children population after mutation, obtaining a new fitness matrix ($\mathbf{F}_{\mathrm{child}}$), which will be the input for the next generation along with $\mathbf{X}_{\mathrm{child}}, \mathbf{Y}_{\mathrm{child}}$.

## 3.5 Hyperparameters tuning

To ensure an effective usage of the GA described in the previous paragraphs, the hyperparameters mentioned in Table 1 have to be properly tuned. The tuning phase presented in this study focuses only on the values of $p_s$, $p_m$, $s_m$ and $d_{\mathrm{lim}}$, whereas $n_{\mathrm{keep}}$ is assumed equal to 3 for each simulation. On the other hand, $n_{\mathrm{gen}}$ and $n_{\mathrm{pop}}$ are chosen depending on the specific analysis since a saturation behaviour is expected instead of finding optimal values.

The concept of dynamic mutation and selection is tested, which consists in changing the hyperparameters depending on the generation number (Hassanat et al., 2019). In this case, the purpose is to foster the mutation in the earliest generations to maxi-

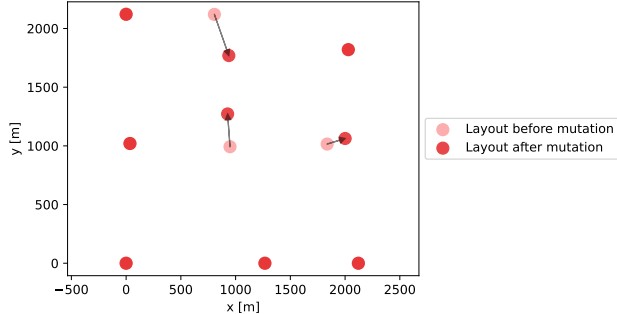

**Figure 4.** Example of mutation. This example does not represent an optimized layout used in this study but it has the only purpose of clarifying the process adopted to apply this technique.

**Table 2.** Optimal values for the hyperparameters of GA. Two numbers are specified, referring to the first and the last generation, respectively. These values are adopted in the simulations performed in this study.

| Hyperparameter | Optimal values |
|---|---|
| $p_s$ | $0.7 \rightarrow 0.7$ |
| $p_m$ | $0.4 \rightarrow 0.05$ |
| $s_m$ | $3\,D \rightarrow 0.2\,D$ |
| $d_{\lim}$ | $2\,D$ |

mize the randomness and adopting a less aggressive mutation in the last generations, where only few refinements are intended. Specifically, the maximum and the minimum values are specified for each parameter to tune and a linear increase/decrease is adopted to obtain the values for each generation. Therefore, different combinations of hyperparameters are tested in order to

find the optimal values for this analysis. The WFLOP described in the base case in section 5.2 is adopted at this stage.

The results of this tuning phase are depicted in Fig. 5, where the violin plots of different cases specified on the x-axis are shown. To limit the computational resources of this phase, when a hyperparameter is tested, the others are kept constant and equal to a pre-defined values. The reason why violin plots are included instead of individual values is related to the random nature of the LO-GA, which requires a statistical interpretation of the results. The values present in the violin plot refer to the

percentage difference in the AEP with respect to the average value of all the simulations performed in this hyperparameter tuning analysis. The optimal values are summarized in Table 2 and are adopted in the other simulations performed in this study. It can be observed that the dynamic mutation is favorable for the performance of the LO-GA whereas a constant value of percentage of selection for all the generations is more effective.

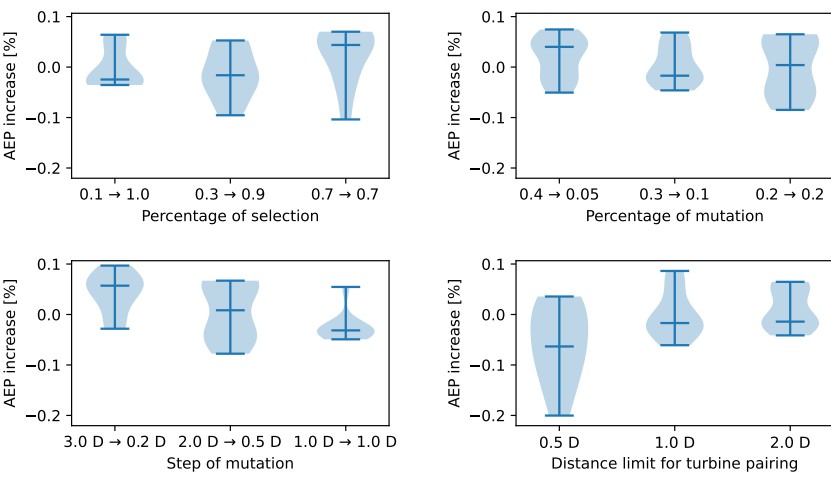

**Figure 5.** Violin plots of the hyperparameter tuning. 10 simulations of the same case are performed to create each plot. Unless explicitly mentioned in the x-axis of each plot, the following hyperparameters are adopted: $p_s = 0.3 \rightarrow 0.9$, $p_m = 0.3 \rightarrow 0.1$, $s_m = 2\,D \rightarrow 0.5\,D$, $d_{\mathrm{lim}} = 2\,D$. The remaining hyperparameters are set as follows: $n_{\mathrm{pop}} = 100$, $n_{\mathrm{gen}} = 100$, $n_{\mathrm{keep}} = 3$.

## 3.6 Multi-objective LO-GA

A multi-objective version of the LO-GA described in the previous sections has been developed in this work to enable a wider evaluation of the co-design approach. Such implementation requires some modifications to be applied to a multi-objective optimization problem and it is based on the concepts of non-domination rank and crowding distance (Deb et al., 2002). As described in the previous sections, the fitness evaluation of every individual turbine is required for the crossover and the mutation phases. Specifically, these values are used to apply the linear crossover technique e to determine the probability of mutation,

whose require the turbines to be ranked based on their fitness value. However, in case of a multi-objective optimization problem, the ranking of the turbines is not univocal. Therefore, to avoid the ambiguity introduced by the multiple objectives, the fitness of the turbines is determined by the norm-1 between the normalized fitness values of the different objectives. This allows to preserve the structure of the crossover and mutation phases described in sections 3.3 and 3.4.

## 4 Geometric yaw relations

This section describes the novel geometric yaw relations developed in this study and adopted for the co-design simulations. First, the dependence on the main variables is discussed and new relations are introduced, comparing these approaches to the work of Stanley et al. (2023). Second, various effects that impact the optimal yaw angle of a turbine are examined and approximated through a new expression that ensures a higher accuracy in optimal yaw predictions.

## 4.1 Dependence on geometric variables

Stanley et al. (2023) have introduced a geometric yaw relation to approximate the optimal yaw angle of a turbine based on the streamwise ($dx$) and the cross-stream ($dy$) distance to its nearest downstream waked turbine. This relation is linear with respect to $dx$ whereas the influence of $dy$ is only limited to the sign of the geometric yaw value, as shown in Eq. 8 (Stanley et al., 2023).

$$\gamma_{\text{geom}} = \text{sign}(dy) \cdot 30 \cdot \left[1 - \frac{1}{25}\frac{dx}{D}\right] \tag{8}$$

However, it can be detected from the study of Stanley et al. (2023) that a decreasing trend is present between the absolute value of $dy$ and the optimal yaw angles. Therefore, two novel relations are introduced to capture this behaviour, included in Eq. 9 and 10, respectively. The first relation extends the linear behaviour on $dx$ present in Eq. 8 also to $dy$ and is characterized by three coefficients ($\gamma_{max}, m_x, m_y$) that have to be properly tuned. This relation is appointed as linear approach in this study. On the other hand, Eq. 10 is based on exponential relations in order to guarantee higher flexibility in the shape of the 275 curves, ensured by two more coefficients to tune. Therefore, this novel expression is referred as exponential approach and is characterized by five coefficients ($\gamma_{max}, p_x, p_y, q_x, q_y$).

$$\gamma_{\text{geom}} = \text{sign}(dy) \cdot \max\left[\gamma_{max} - m_x\frac{dx}{D} - m_y\frac{|dy|}{D}, 0\right] \tag{9}$$

$$\gamma_{\text{geom}} = \text{sign}(dy) \cdot \gamma_{max} \cdot \frac{p_x + 1}{p_x + \exp\left(\frac{1}{q_x}\frac{dx}{D}\right)} \cdot \frac{p_y + 1}{p_y + \exp\left(\frac{1}{q_y}\frac{|dy|}{D}\right)} \tag{10}$$

These approaches described in Eq. 9 and 10 focus only on the position of the nearest downstream waked turbine for the 280 calculation of the geometric yaw, whose identification is performed as described by Stanley et al. (2023). However, it is preceded by a filtering phase based on the effective wind speed value assuming that wake steering is not applied. Specifically, the effective wind speed ($ws_{\text{eff}}$) is computed for each wind turbine of the farm using the appropriate PyWake function. Then, the turbines characterized by $ws_{\text{eff}} > ws_{\text{rated}}$ are filtered out from the identification of the nearest waked turbine. This stage is introduced to avoid that a loss in power in the upstream turbine when an increase in the wind speed experienced by the 285 downstream turbine does not affect its power generation, i.e. it is operating above rated region. This step is referred as the wind speed filtering phase and slightly increases the computational time of the process. However, a gain in accuracy of the geometric yaw approximation is expected.

## 4.2 Implementing corrections to the geometric yaw

The purpose of this section is to apply different corrections to the geometric yaw relation, aiming to improve the approximation 290 of the optimal yaw angle. The novelty of this method is based on considering multiple waked turbines instead of limiting the

relation to the nearest turbine. This new approach is referred as exponential corrected relation since it is based on the Eq. 10. The corrective/filtering steps applied in this approach are summarized in the following list and explained in detail in the next paragraphs.

1. Wind speed filtering

2. Initial optimal yaw approximation

3. Optimal yaw correction

4. Effective wind speed correction

The wind speed filtering phase consists in the calculation of the effective wind speed assuming no wind farm control strategy. However, in this case not only the turbines characterized by $ws_{\mathrm{eff}} > ws_{\mathrm{rated}}$ are filtered out from the identification of the waked turbines, but also those presenting $ws_{\mathrm{eff}} < ws_{\mathrm{cut-in}} - \delta_{\mathrm{cut-in}}$. In this study, $\delta_{\mathrm{cut-in}} = 2$ ms$^{-1}$ is assumed. The reasoning behind this additional correction is to avoid the penalization of the upstream turbine caused by the yaw misalignment if the gain in wind speed for the downstream turbine is not sufficient to achieve a value higher than $ws_{\mathrm{cut-in}}$.

After the wind speed filtering, an initial approximation of the optimal yaw angle ($\gamma_{\mathrm{initial}}$) is calculated based on Eq. 10. Unlike the previous approaches focused only on the nearest waked turbine, the geometric yaw is calculated considering every downstream waked turbine individually, identified through the variables $dx$ and $dy$. Among these multiple geometric yaw values, the $\gamma_{\mathrm{initial}}$ is given by highest of these values. Even though in many cases this value is determined by the nearest downstream waked turbine, there are some situations when such turbine does not coincide with the one that influences the most the optimal yaw of the upstream turbine. This is explained through an example depicted in Fig. 6. In this illustration, three turbines are considered, and the optimal yaw angle of the first turbine is studied in relation to the position of the third turbine. These optimal yaw angles depicted in the figure on the right are calculated through the serial-refine method, hence they represent the target for the geometric yaw relation. The dashed line represents the optimal yaw angle of the first turbine in case only the second turbine is present, i.e. the nearest turbine. It can be observed that for high values of $dy$, the third turbine does not influence the optimal yaw angle of the first turbine. However, in case of a better alignment with the first turbine, i.e. $dy \approx 0$, the third turbine has an impact on the value even though it is not the nearest turbine. In summary, this initial yaw approximation identifies the downstream waked turbine that has the highest impact on the optimal yaw angle and calculates the geometric yaw based on the $dx$ and $dy$ of this turbine.

The initial yaw approximation is then corrected considering the influence of the other downstream waked turbines. The aim of this correction is to avoid that the wake of the upstream turbine is steered towards other downstream turbines after the initial yaw approximation. Suppose that the most impactful downstream turbine on the optimal yaw angle of the upstream turbine is characterized by $\mathrm{sign}(dy) > 0$. The initial yaw approximation causes a steering of the wake towards the region such that $\mathrm{sign}(dy) < 0$. Therefore, only the turbines characterized by a $\mathrm{sign}(dy)$ opposite to the $\mathrm{sign}(dy)$ of the turbine that has determined the initial yaw approximation are relevant for this correction. Similarly to the previous case, an example is introduced to clarify the need of this correction, illustrated in Fig. 7. It can be observed that the third turbine has no influence

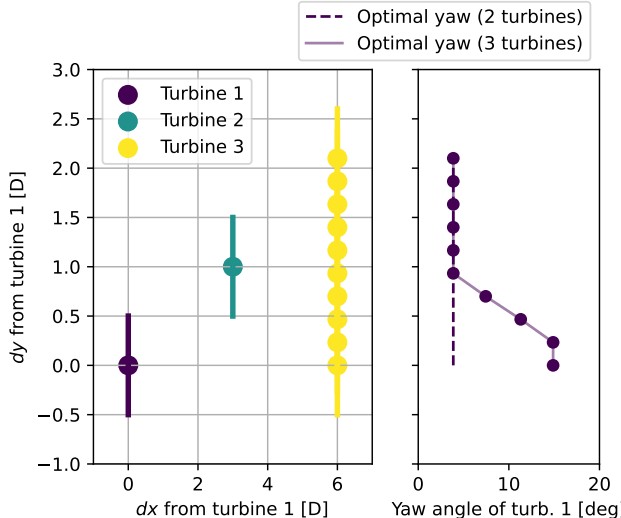

**Figure 6.** Example introduced to explain the reasoning behind the initial optimal yaw approximation. The influence on the optimal yaw angle of the upstream turbine is studied depending on the positions of two downstream turbines. The optimal yaw angle of turbine 1 is studied in relation to the position the turbine 3. The left plot illustrates the different positions considered in this example, The right plot shows the optimal yaw angle of the turbine 1 for two cases: neglecting the presence of turbine 3 (dashed line) and considering different positions (identified by $dy$) of turbine 3 (solid line).

on the optimal yaw angle of the first turbine in case of $dy \ll 0$. However, as the $dy$ of the third turbine increases, the optimal yaw angle decreases to avoid that the wake is steered toward this turbine. However, if $dy \approx 0$ it becomes more convenient to increase the magnitude of the yaw of the first turbine in order to steer the wake away from both turbines.

This situation is tackled as follows. First, the wake deflection caused by $\gamma_{\text{initial}}$ is calculated using the approach of Jiménez et al. (2010). Therefore, the local change in wind direction experienced by the downstream waked turbine is calculated through Eq. 11 (Jiménez et al., 2010). $C_T$ and $k$ refer to the thrust coefficient and the wake expansion coefficient, respectively. The former is extracted from the turbine data, depending on the free-stream wind speed, whereas the latter is assumed equal to $0.1$.

$$\delta_{wd} = -\frac{C_T}{2} \frac{\sin(\gamma_{\text{initial}}) \cdot \cos^2(\gamma_{\text{initial}})}{1 + k \cdot \frac{dx}{D}} \tag{11}$$

The values of $dx$ and $dy$ of the turbine of interest, i.e. such that $\text{sign}(dy)$ is opposite to the value from which $\gamma_{\text{initial}}$ is calculated, are then modified through a rotation of $\delta_{wd}$. Therefore, new values $dx_{\delta_{wd}}$ and $dy_{\delta_{wd}}$ are obtained, describing the position of the relevant downstream waked turbines based on the direction of the deflected wake. Subsequently, the method used to calculate $\gamma_{\text{initial}}$ is repeated using $dx_{\delta_{wd}}$ and $dy_{\delta_{wd}}$ as input for the exponential relation of Eq. 10. Therefore, a new geometric yaw value for the upstream turbine is obtained, appointed as geometric yaw correction ($\gamma_{\text{corr}}$) and used to improve the initial approximation as described in Eq. 12. The coefficient $\alpha_{\text{corr}}$ is properly tuned to weight this yaw correction accordingly.

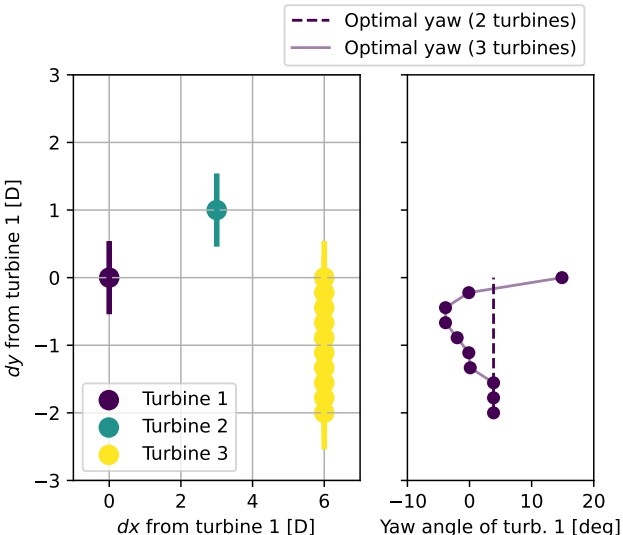

**Figure 7.** Example introduced to explain the reasoning behind the optimal yaw correction. The influence on the optimal yaw angle of the upstream turbine is studied depending on the positions of two downstream turbines. The optimal yaw angle of turbine 1 is studied in relation to the position the turbine 3. The left plot illustrates the different positions considered in this example, The right plot shows the optimal yaw angle of the turbine 1 for two cases: neglecting the presence of turbine 3 (dashed line) and considering different positions (identified by $dy$) of turbine 3 (solid line).

$$\gamma_{\text{geom}} = \gamma_{\text{initial}} + \alpha_{\text{corr}} \cdot \gamma_{\text{corr}} \tag{12}$$

Lastly, a wind speed correction factor ($f_{ws}$) is multiplied to the geometric yaw to penalize high values of $\gamma_{\text{geom}}$ in case the deviation of the effective wind speed from the free-stream value is limited. The motivation to include such correction in the geometric yaw is related to cubic dependence of the generated power with respect to the incident wind speed. $f_{ws}$ is calculated as described in Eq. 13, where the coefficient $\alpha_{ws}$ is part of the tuning process.

$$f_{ws} = 1 - \alpha_{ws} \cdot \exp\left(ws_{\text{eff}} - ws\right) \tag{13}$$

Table 3 summarizes the geometric yaw relations analysed in this study along with their coefficients, which are properly tuned as described in the next section.

### 4.3 Tuning of the coefficients

The tuning of the coefficients of the relations introduced in the previous paragraphs aims to guarantee a precise approximation of the optimal yaw angles. The method adopted for this tuning phase assumes that the optimal values of the coefficients are

**Table 3.** Geometric yaw relations analysed in this study.

| Name | Relation | Coefficients |
|---|---|---|
| Stanley relation | $\gamma_{\text{geom}} = \text{sign}(dy) \cdot 30 \cdot \left[1 - \frac{1}{25}\frac{dx}{D}\right]$ | - |
| Linear relation | $\gamma_{\text{geom}} = \text{sign}(dy) \cdot \max\left[\gamma_{max} - m_x\frac{dx}{D} - m_y\frac{|dy|}{D}, 0\right]$ | $\gamma_{max}, m_x, m_y$ |
| Exponential relation | $\gamma_{\text{geom}} = \text{sign}(dy) \cdot \gamma_{max} \cdot \frac{p_x+1}{p_x+\exp\left(\frac{1}{q_x}\frac{dx}{D}\right)} \cdot \frac{p_y+1}{p_y+\exp\left(\frac{1}{q_y}\frac{|dy|}{D}\right)}$ | $\gamma_{max}, p_x, p_y, q_x, q_y$ |
| Exponential corrected relation | $\gamma_{\text{geom}} = f_{ws} \cdot (\gamma_{initial} + \alpha_{\text{corr}} \cdot \gamma_{\text{corr}})$ | $\gamma_{max}, p_x, p_y, q_x, q_y, \alpha_{\text{corr}}, \alpha_{ws}$ |

those that lead to the maximization of the AEP when the geometric yaw relations are applied. Specifically, the tuning of the
350 coefficients can be divided into two different steps. First, different wind farm layouts are generated to evaluate the geometric
yaw relations. Second, an optimization problem is solved to extract the values of the coefficients that maximize the AEP for
the given layouts.

Since in the second step the AEP is calculated for all the layouts at each evaluation of the objective function, the total number
of layouts has to be limited to avoid excessive computational requirements. On the other hand, these layouts have to be chosen
in order to prevent that the geometric yaw relations are effective only for few specific cases. Therefore, 20 different layouts are
generated through few generations of the LO-GA. Specifically, population size and number of generations are set equal to 100
and 10, respectively, whereas the number of turbines and the power density are selected randomly in the ranges $16 - 72$ and
$10 - 20\,\text{W}\,\text{m}^{-2}$.

After the generation of these layouts, the optimization problem to tune the coefficients of geometric yaw relations is struc-
360 tured as follows. The objective function consists in the average percentage increase in the AEP among the 20 different layouts
when applying the geometric yaw relation with respect to the case without wake steering. The optimization variables are the
coefficients of the geometric yaw relations, which are bounded in ranges determined by the experience based on preliminary
simulations. This optimization problem is solved using a basic GA implementation in PyGad ($n_{\text{pop}} = 50$, $n_{\text{gen}} = 30$). The
results of this tuning phase, i.e. the optimal coefficients, are included in Table 4 and adopted during the simulations performed
in this study.

Among the various coefficients present in the geometric yaw relations and included in Table 4, $\gamma_{max}$ has a clear physical
interpretation since it represents the maximum absolute value for the yaw angles of the turbines. In this study, $\gamma_{max}$ has
been tuned targeting the maximization of the AEP without applying any restriction. However, the tuning of $\gamma_{max}$ could be
constrained to take into account actuation limits or requirements on the structural loading, since it has been demonstrated that
wake steering can have a negative impact on some load channels (Shaler et al., 2022).

## 5 Case study

This section defines the case study adopted in this work to evaluate the co-design approach for the WFLOP. First, the wind
scenario and the turbines adopted in the simulations are defined. Second, the base-case selected to evaluate the different

**Table 4.** Tuned coefficients of the geometric yaw relations.

| Stanley relation | | Linear relation | | Exponential relation | | Exponential corrected relation | |
|---|---|---|---|---|---|---|---|
| - | - | $\gamma_{max}$ | 19.788° | $\gamma_{max}$ | 20.928° | $\gamma_{max}$ | 20.771° |
| | | $m_x$ | 0.424 | $p_x$ | 8.146 | $p_y$ | 5.069 |
| | | $m_y$ | 12.019 | $p_y$ | 6.320 | $p_y$ | 7.474 |
| | | | | $q_x$ | 5.381 | $q_x$ | 6.519 |
| | | | | $q_y$ | 0.346 | $q_y$ | 0.393 |
| | | | | | | $\alpha_{\text{corr}}$ | 0.473 |
| | | | | | | $\alpha_{ws}$ | 0.091 |

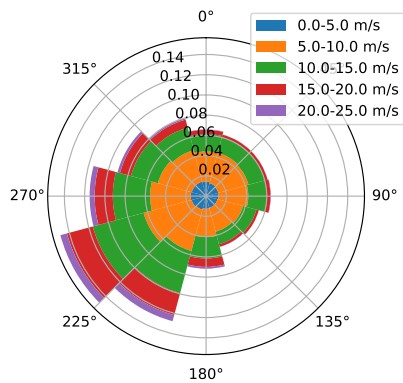

**Figure 8.** Wind rose of Hollandse Kust Noord site.

geometric yaw relations is introduced. Lastly, the modifications introduced to the base-case to perform a sensitivity analysis are explained.

## 5.1 Site and turbines

The wind conditions adopted for the simulations in this study refer to the Hollandse Kust Noord (HKN) site in the Netherlands (Netherlands Enterprise Agency, 2019). The wind rose is computed assuming a uniform Weibull distribution from the scale and shape factor of 12 sectors, as shown in Fig. 8. The turbine type chosen for the simulation is the reference DTU 10MW, available in PyWake.

## 5.2 Base case

The aim of this base-case is to evaluate the potential of the co-design approach in comparison with the study of Stanley et al. (2023), assuming HKN site conditions. Therefore, a 16-turbine wind farm is adopted, characterized by a power density of $20\,\text{Wm}^{-2}$ and a squared available area. 36 and 23 bins are used during the simulations for the wind direction and the wind

**Table 5.** Base-case

| | |
|---|---|
| Wind rose | HKN |
| Turbine | DTU 10 MW |
| Wind speed bins | 23 |
| Wind direction bins | 36 |
| Minimum distance constraint | $2\,D$ |
| Power density | $20\,\mathrm{Wm}^{-2}$ |

speed, respectively. Lastly, the minimum distance constraint ($d_{\min}$) is set equal to $2\,D$. These information are summarized in Table 5.

## 5.3   Sensitivity analysis

Due to the limited size and the large power density of the farm studied in the base-case, a sensitivity analysis is performed to understand the potential of the co-design approach for a wider range of conditions. Specifically, different power density

values are tested to match conditions similar to current wind farm development projects. This is achieved by keeping the same number of turbines included in the base-case while extending the available area. On the other hand, larger size of wind farms are simulated to investigate the benefits of this method for future large plants. Therefore, new simulations are performed increasing both the number of turbines and the available area, while keeping the power density equal to $20\,\mathrm{Wm}^{-2}$. Lastly, the impact of the site is analyzed in terms of distribution of the wind probability among the different wind sectors. For this

purpose, average scale and shape factors of HKN site are assumed while the probability of occurrence of each wind direction is modified. Specifically, this is modelled as a Gaussian distribution and different values for the standard deviation are used. Adopting this method, various conditions are obtained ranging from unidirectional wind roses to omnidirectional cases. These are depicted in Fig. 9, where the HKN site is included as well. In general, the same conditions of the base case are adopted in the sensitivity analysis, unless explicitly specified.

## 6   Results

This section illustrates the results of the simulations performed in this analysis to evaluate the potential of the co-design approach for WFLOP. First, the LO-GA is evaluated in comparison to a general GA implementation. Second, the geometric yaw relations introduced in this work are assessed through a preliminary test and the results obtained in the base-case simulations are shown. Then, the influence of the power density, the number of turbines and the site type is studied through a sensitivity

analysis. Lastly, the results concerning a multi-objective implementation of the co-design approach are shown and the impact of the wind direction discretizatiotn is investigated.

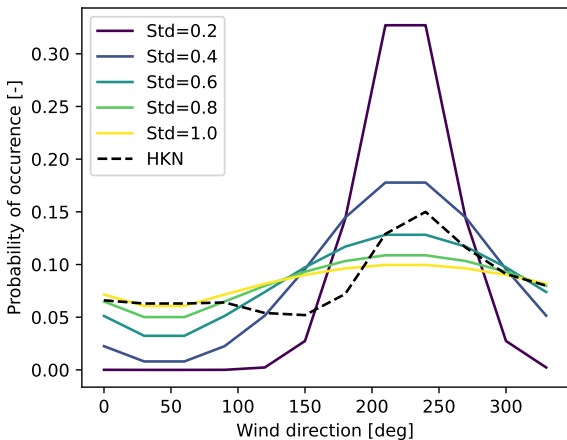

**Figure 9.** Probability of occurrence of different wind directions for the cases studied in the sensitivity analysis. HKN values are included and indicated with the dashed line.

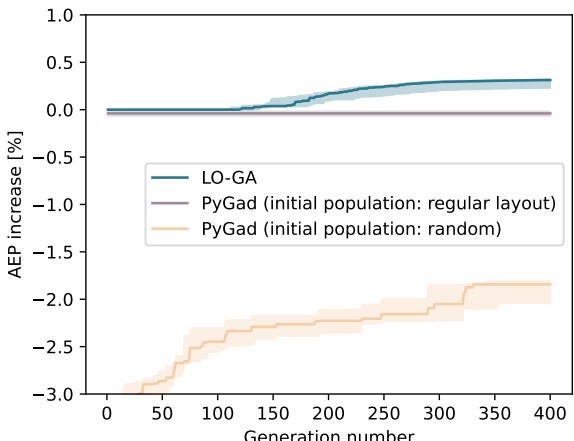

**Figure 10.** Comparison between LO-GA and PyGad GA to solve the WFLOP. The solid lines represent the median AEP increase with respect to the regular layout, whereas the areas refer to the range between the $25^{th}$ and $75^{th}$ percentiles, being the results of multiple simulations of the same case.

## 6.1 Evaluation of the genetic algorithm

The LO-GA is compared to the basic implementation of the GA in PyGad, introduced in Fig. 1. The results are depicted in Fig. 10, where it can be observed that the LO-GA outperforms the PyGad implementation, being able to improve the regular layout from which the population is initialized.

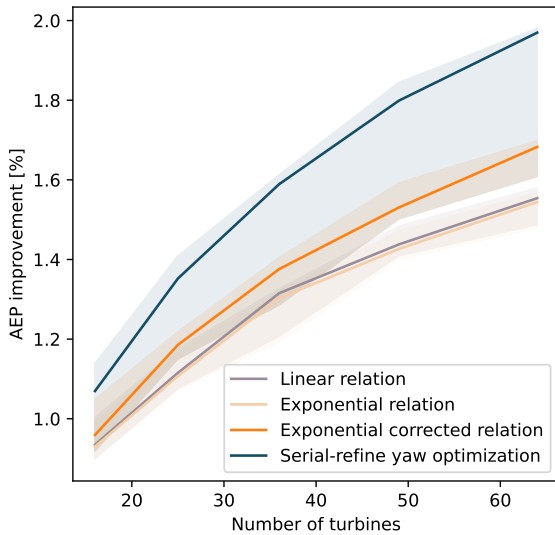

**Figure 11.** Preliminary test to evaluate the geometric yaw relations. The solid lines represent the median values whereas the area refer to the range between the $25^{\text{th}}$ and $75^{\text{th}}$ percentiles, being the results of multiple simulations of the same case.

## 6.2 Preliminary test on geometric yaw relations

To evaluate the effectiveness of the geometric yaw relations summarized in Table 3, a preliminary analysis is performed and the results are depicted in Fig. 11. Specifically, for different farm sizes, the percentage difference in the AEP is calculated between the case when the geometric yaw angles are applied and the case when wake steering is not considered. This is compared
also with the AEP value obtained from the yaw angles computed through the serial-refine method (Fleming et al., 2022), considered the optimal yaw angles in this study. For each size of the farm, this calculation is performed on 10 different layouts optimized using the LO-GA ($n_{\text{pop}} = 100$, $n_{\text{gen}} = 200$), assuming a power density of $20\,\text{W}\text{m}^{-2}$. It can be observed that the linear and the exponential relation guarantee approximately the same increase in the AEP whereas higher values are obtained for the exponential corrected relation. This indicates that the corrections included in this approach improve the optimal yaw
estimation. The results concerning the Stanley relation are not included in the figure since significantly lower values of AEP increase are produced.

## 6.3 Base-case

The results obtained for the base-case are included in Fig. 12, where the procedure explained in section 2.2 is followed, namely comparing optimized layouts resulted from including or neglecting the geometric yaw angles within the objective function of
the WFLOP. In both cases, the optimal yaw angles are then applied to calculate the actual AEP associated to the layout. Due

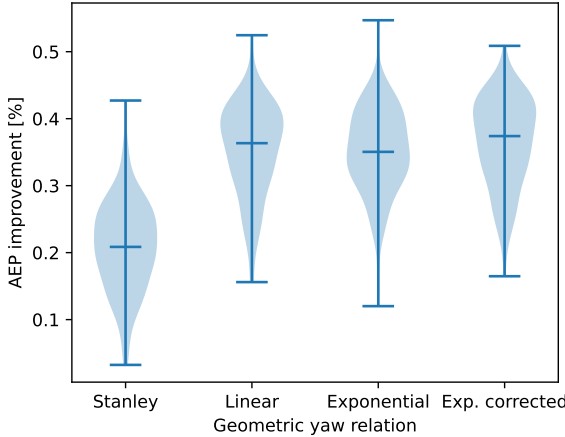

**Figure 12.** AEP improvement obtained with co-design approach for WFLOP using different geometric yaw relations. The distributions of multiple iterations for each case are included, highlighting the median value.

to the random nature of the LO-GA, 100 iterations of each case are performed, obtaining the distributions shown in the plots. In this case, the hyperparameters adopted for the layout optimizations are $n_{\mathrm{pop}} = 100$ and $n_{\mathrm{gen}} = 400$. It is evident that the relations introduced in this paper, i.e. linear, exponential and exponential corrected, outperform the Stanley relation. However, no significant difference can be detected among the three new approaches. Specifically, the higher gains of the exponential corrected relation observed in Fig. 11 do not guarantee significantly better performance for the co-design method. Overall, considering median values, an AEP increase between $0.3\%$ and $0.4\%$ is obtained for this base-case.

The improvements in the AEP obtained with the co-design approach for WFLOP can be achieved only if wake steering is applied during the operation of the farm. Therefore, in case wake steering is not adopted, a drop in performance is experienced for the layouts optimized using the co-design method. Assuming no wake steering during the operation, Fig. 13 illustrates the reduction in the AEP with respect to layouts optimized with the traditional approach. These results are generated using the same method and parameters of Fig. 12. From this analysis it can be concluded that the drop in performance has the same magnitude and follows the same trend of the gains shown in Fig. 12.

### 6.4 Sensitivity analysis

This section presents the results concerning the sensitivity analysis on the co-design approach in relation with three different aspects: power density, farm size and site type.

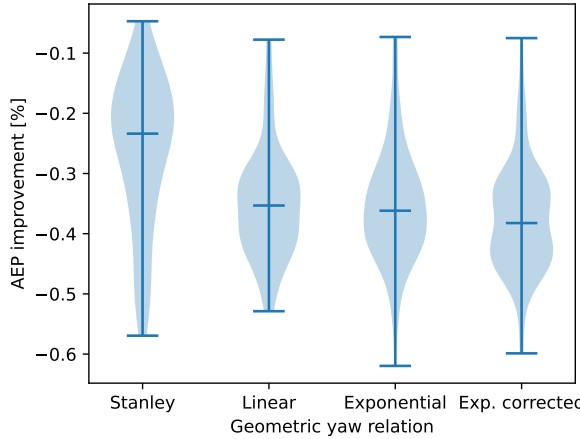

**Figure 13.** AEP reduction obtained with co-design approach for WFLOP using different geometric yaw relations in case wake steering is not applied during the operation. The distributions of multiple iterations for each case are included, highlighting the median value.

### 6.4.1 Power density

The AEP increase obtained through the co-design approach is dependent on the power density of the wind farm, as highlighted by the solid lines in Fig. 14. Specifically, it can be observed that the benefits of this method diminish for values below $15\,\mathrm{Wm^{-2}}$, representing a threshold after which the gain stabilizes. These results demonstrate the effectiveness of the co-design method only for wind farms constrained in a limited area, i.e. characterized by a higher power density. For this analysis the optimization hyperparameters are set to $n_{\mathrm{pop}} = 100$ and $n_{\mathrm{gen}} = 400$ and the randomness associated to the results is handled by computing each case 25 times. Similarly to the base case, the drop in performance in case wake steering is not applied during the operation is investigated for different power densities. These results are included in Fig. 14 using dashed lines, exhibiting an opposite trend with respect to the case when wake steering is applied, as observed for the base case. However, the saturation behavior observed for values higher than $15\,\mathrm{Wm^{-2}}$ is not present, since the performance worsen for larger values.

Further analysis is performed to investigate whether the saturation limit of $15\,\mathrm{Wm^{-2}}$ is dependent on the squared shape of the domain or it remains valid for other geometries. For this purpose, the simulations are repeated for two different rectangular areas, characterized by a ratio between the sides equal to $1.5$ and $2.0$, respectively. The results are limited to the exponential corrected relation and are included in Fig. 15. It can be observed that the magnitude of the AEP gains decreases and the saturation behavior identified in the previous case is not evident anymore, concluding that such results are dependent on the shape of the area where the turbines can be positioned. However, the decreasing trend for the AEP gains in case of lower power density values remains valid, hence it can be considered a general conclusion for this sensitivity analysis.

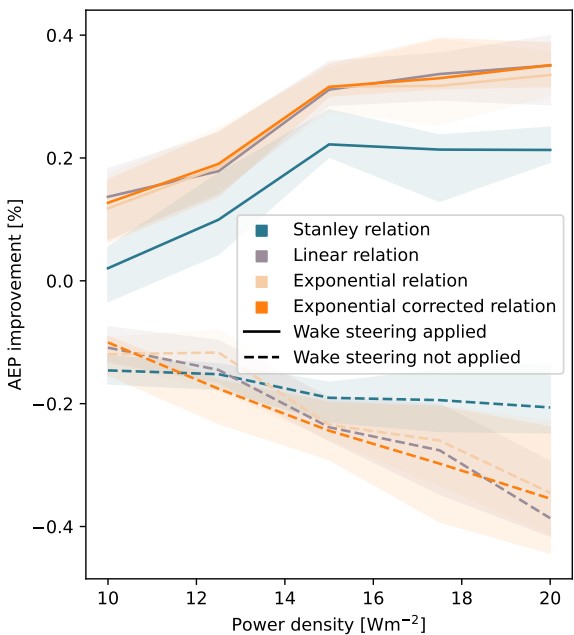

**Figure 14.** AEP gain obtained with co-design method for different power densities, showing the cases when wake steering is applied (solid lines) and not applied (dashed lines) during the operation of the farm. The areas surrounding the lines (median values) refer to the range between the $25^{\text{th}}$ and $75^{\text{th}}$ percentiles, being the results of multiple simulations of the same case.

### 6.4.2 Number of turbines

The co-design approach is tested for different farm sizes to understand its potential if the number of turbines increases. The results are included in Fig. 16, where each case is simulated 20 times, adopting $n_{\text{pop}} = 100$ and $n_{\text{gen}} = 600$ as optimization hyperparameters. The choice of a higher number of generations is needed to ensure the convergence when the number of optimization variables increases. In this case, it can be observed that the AEP improvement increases up to $0.6\%$ and a saturation behavior can be detected when $n_{wt} > 25$. However, as $n_{wt}$ increases, the trend becomes less evident due to the higher oscillations between different runs, caused by the limited hyperparameters values in proportion to the number of optimization variables. This occurs also in the case when the robustness of the method is tested when wake steering in not applied for a co-design optimized layout, as observed form the dashed lines in Fig. 16.

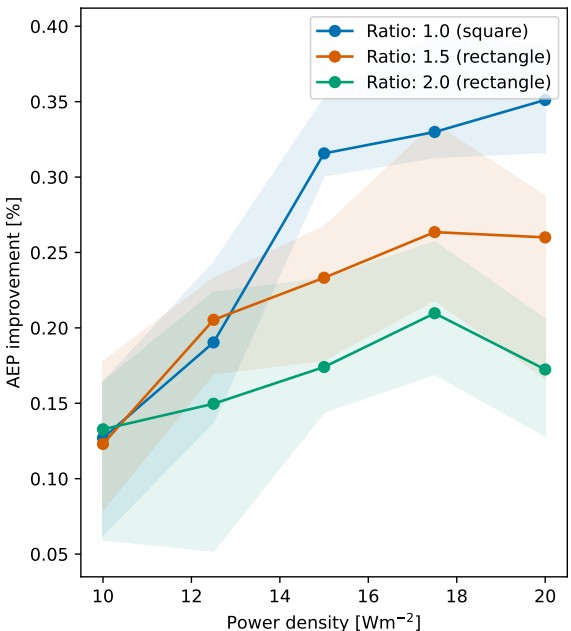

**Figure 15.** AEP gain obtained with co-design method (exponential corrected relation) for different power densities, showing the trend for different shapes of the available surface, identified through the ratio between the sides of a rectangle. The areas surrounding the lines (median values) refer to the range between the $25^{\text{th}}$ and $75^{\text{th}}$ percentiles, being the results of multiple simulations of the same case.

### 6.4.3 Wind direction variability

Various site types are investigated to determine the effectiveness of the co-design method in different wind conditions. As mentioned in section 5.3, this study focuses on the shape of the wind rose in terms of the probability of occurrence for the wind directions. Figure 17 shows that the co-design approach becomes less convenient for sites characterized by an evident dominant wind direction, i.e. low standard deviation of the probability of occurrence along the $360°$. This can be observed not only by a drop in the increase in AEP when wake steering is applied during the operation, but also by a significant reduction in case wake steering is not adopted. On the other hand, the increase in the AEP does not vary for values of standard deviations higher than $0.4$. Similarly to the power density, these results have been obtained from 25 iterations, adopting $n_{\text{pop}} = 100$ and $n_{\text{gen}} = 400$.

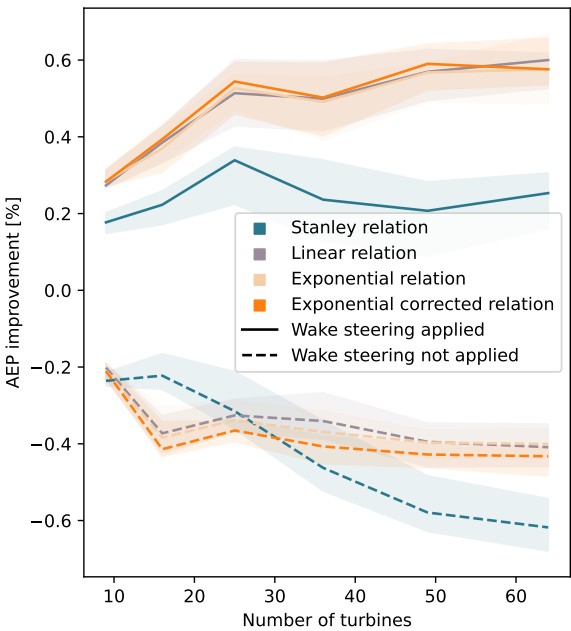

**Figure 16.** AEP gain obtained with co-design method for farm sizes, showing the cases when wake steering is applied (solid lines) and not applied (dashed lines) during the operation of the farm. The areas surrounding the lines (median values) refer to the range between the $25^{\text{th}}$ and $75^{\text{th}}$ percentiles, being the results of multiple simulations of the same case.

## 6.5 Multi-objective co-design approach

The previous results have highlighted the negative impact in term of AEP losses in case wake steering is not applied during the operation phase in a layout optimized through the co-design approach. This is tackled by performing a multi-objective optimization that maximizes the AEP for both the case when wake steering is applied and when this does not occur during operation. This analysis is limited to the base case and to the exponential corrected geometric yaw relation, and it is performed using the multi-objective version of the LO-GA introduced in section 3.6. The results are outlined in Fig. 18, where the Pareto front is visible on the top-right, resulting from a max-max optimization. These have been obtained after 50 iterations of each case, for which the hyperparemeters adopted for the base case are used. As highlighted by the cross in Fig. 18, it can be extracted from the Pareto front a layout that limits the AEP losses to $0.1\%$ in case wake steering is not applied while ensuring a gain higher than $0.3\%$ if wake steering takes place.

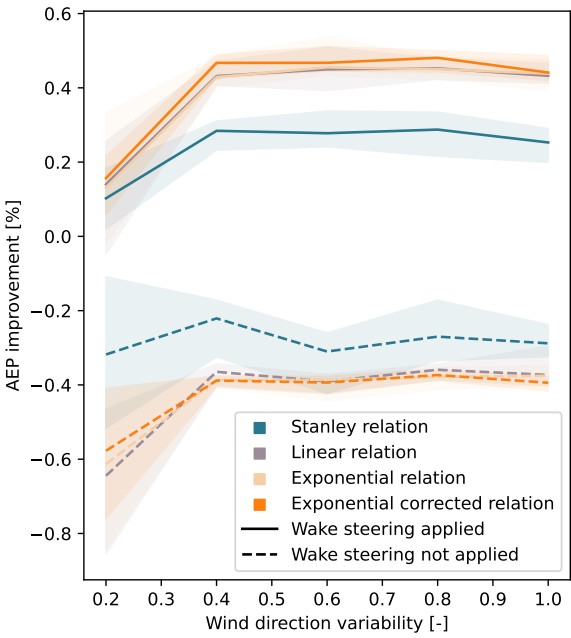

**Figure 17.** AEP gain obtained with co-design method for site types, showing the cases when wake steering is applied (solid lines) and not applied (dashed lines) during the operation of the farm. The x-axis refers to the standard deviation of the probability of occurrence of each wind direction, i.e. from unidirectional (low values) to omnidirectional (high values) wind roses. The areas surrounding the lines (median values) refer to the range between the $25^{th}$ and $75^{th}$ percentiles, being the results of multiple simulations of the same case.

## 6.6 Effect of wind direction discretization

The AEP calculation within this study is based on the traditional procedure of discretizing the wind direction, selecting appropriate bins at which the power production of the farm is computed. The resolution of this discretization can have a significant impact on the WFLOP since a sufficiently high resolution is required to consider all the wake interactions between the turbines. On the other hand, using a fine resolution for the wind directions can increase significantly the computational cost of the optimization. As mentioned in section 5.2, 36 wind direction bins are adopted in our simulations, namely the wind direction is discretized using bins of $10°$. The purpose of this section is to investigate the influence of this parameter on the co-design approach described in this study.

The simulations of the base-case are repeated modifying the size of the bins adopted for the wind direction, hence altering the objective function used to calculate the AEP. Since such function is used also during the tuning phase of the LO-GA hyperparameters, their values are recalculated accordingly. Specifically, a higher randomization of the initial population has shown to be beneficial for both the sequential and the co-design approach, obtaining higher AEP values. This is applied for

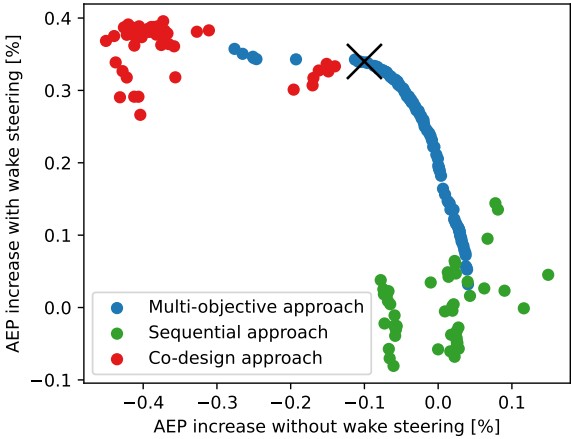

**Figure 18.** Multi-objective co-design approach based on the multi-objective optimization. Each data point indicates an optimized layout. Multiple data points are included also for the sequential and the co-design approach, resulting from different simulations and providing a probabilistic interpretation of the results in accordance with the other graphs. The axis of the plot refer to the AEP increase with respect to the average AEP value obtained through the traditional sequential approach. The Pareto front is present on the top-right of the plot, resulting from a max-max problem. A cross is included to identify a possible robust solution.

all the wind direction resolutions tested in this section to provide consistency between the results. Moreover, for each case the same resolution is applied for the layout optimization and the subsequent yaw optimization performed with the serial-refine method.

The results are presented in Fig. 19, which shows a significant drop in the AEP improvement obtained through the co-design approach as the wind direction resolution is increased, i.e the size of the bins becomes smaller. Moreover, the AEP values correspondent the sequential and the co-design layouts are also shown individually to gain further insights on this behavior. These results are limited to exponential corrected relation since the influence of the wind direction resolution does not vary for the different geometric yaw approaches. In general, the values of AEP improvement obtained for $10°$ bin size are lower with respect to the results presented in Fig. 12, as a consequence of the higher randomization of the initial population.

Two different effects can be detected from this analysis. First, the AEP values decrease for both the sequential and the co-design methods when the bin size is halved from $10°$ to $5°$. Increasing the wind direction resolution, more wake interactions are simulated within the farm, amplifying the wake losses. This sensitivity on the wind direction resolution is enhanced by the characteristics of the base-case, where the limited number of turbines and the higher power density lead to fewer wake interactions and prevent wake expansions. Moreover, the squared shape of the domain causes the wind directions that are multiples of $45°$ to be characterized by high wake losses. This occurs due to the tendency of the turbines to be positioned at the corners of the domain to maximize the use of the available area. All of these directions cannot be simulated in case the bin size is set to $10°$, limiting the accuracy of the results. However, the AEP reduction obtained for the $5°$ bin size is

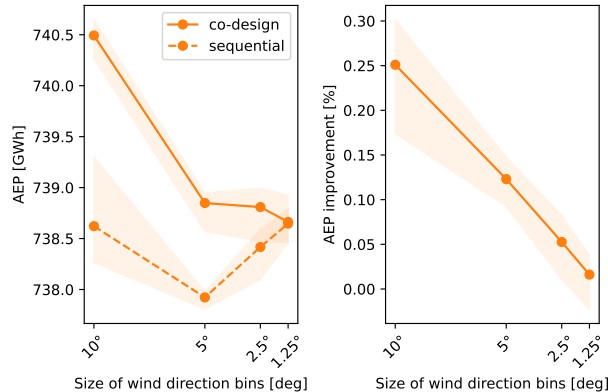

**Figure 19.** Effect of wind direction resolution to the co-design approach, in terms of AEP for the sequential and the co-design layouts (on the left) and AEP improvement (on the right). The areas surrounding the lines (median values) refer to the range between the $25^{\text{th}}$ and $75^{\text{th}}$ percentiles, being the results of multiple simulations of the same case.

more evident for the co-design case, diminishing the improvement obtained through this approach. Nevertheless, this effect is expected to decrease when the number of turbines is larger, due to the occurrence of wake interaction for a wider range of wind directions. The second effect that can be detected from Fig. 19 concerns the AEP trends as the wind direction bin size is reduced from $5°$ to lower values. It can be observed that AEP values of the co-design layouts remain stable whereas those of the sequential layouts increase when a finer resolution is adopted. Therefore, they tend to converge towards the AEP values of the layouts optimized using the co-design method, i.e. leading to an AEP improvement equal to $0\%$ and nullifying the usefulness of this approach. However, such result is highly conditioned by the application of the serial-refine method for yaw optimization with a fine wind direction resolution. For instance, when a bin size of $1.25°$ is adopted for the simulations, the optimal yaw angle is recalculated for every $1.25°$. Therefore, it is assumed that during the farm operation the wind direction is measured with an accuracy higher than $1.25°$ and the yaw angles of turbines are modified every time such change of $1.25°$ is detected. These conditions are far from being realistic, as mentioned by Quick et al. (2020). Therefore, the results presented in Fig. 19 are altered by this assumption which is not consistent with current wind farm operational limits. Nevertheless, it can be concluded that the application of an extremely precise wake steering control would saturate the benefits of the co-design approach proposed in this study.

## 7 Discussion

This section aims to interpret and provide further explanation on the results presented in the previous paragraphs. In general, the methodology introduced in this work to solve the WFLOP based on co-design approach, namely the LO-GA and the different geometric yaw relations, have succeeded to improve the methods available in literature. However, the increase in the AEP of $0.8\%$ described by Stanley et al. (2023) for a site characterized by Gaussian hill spatially varying inflow cannot be

obtained for common sites such as the HKN location adopted for this study. Specifically, values up to $0.6\%$ have been obtained in this analysis, when farms composed by more than 25 wind turbines are considered. As mentioned by Stanley et al. (2023), this limited percentage can be translated into significant amounts in term of revenues and energy production. Specifically, since HKN wind farm plans to fulfill the energy demand equivalent to 1 million households, this increase of $0.6\%$ can be quantified as the consumption of 6 thousand households in this case (Crosswind, 2024). Most importantly, this approach based on geometric yaw angles does not involve significantly higher computational cost with respect to traditional methods, unlike other co-design implementations. On the other hand, the limited improvement in the AEP obtained with the co-design approach is affected by the uncertainty related to the engineering models adopted for the calculations. Moreover, this modest increase in the AEP can be difficult to detect during wind tunnel experiments or field tests. These considerations can challenge the reliability of this method despite the promising results obtained from the simulations.

Besides the benefits associated to the co-design approach in case wind farm control is applied during the operation, this study has also highlighted the downsides of not implementing such control strategy. A decrease in the AEP is registered in this case, as a consequence of adapting the objective function of the WFLOP to the wake steering technique. The magnitude of such loss in the energy production is similar to the gain observed if the coherent procedure is applied. Therefore, it is essential that the wind farm operator takes firm decisions on the control strategy prior to the design phase. Otherwise, the limited improvement of the co-design approach would be further challenged by the uncertainty caused by the future decisions of the wind farm operator. To prevent such situation, a multi-objective co-design approach is proposed in this study, optimizing the wind farm layout for both the case when wake steering is applied and the case when this does not happen. Therefore, such method increases the reliability of the wind farm layout, minimizing the risk related to future decision on the control strategies taken during the operation of the farm.

The perspective and the limitations of the co-design approach for wake steering have been shown by the sensitivity analysis performed in this study. First, a decrease in the AEP gain is observed for low power densities, irrespective of the shape of the available surface. Such behavior proves that including the geometric yaw within the WFLOP is more effective when wake steering plays a major role. In particular, higher gains are obtained when the distances between the turbines decrease hence the wake effect is more impactful. However, after a certain power density this improvement can saturate due to the impossibility of further mitigation of the wake deficit. Second, similar explanations can be provided to justify the dependence of the AEP improvement on the farm size, expressed in term of number of turbines. In principle, extending the size of the farm increases the number of downstream turbines that experience lower wind speeds, hence boosting the potential of wake steering and consequently of the co-design approach. However, a saturation trend can be observed also in this case, caused by the impossibility of deflecting the wake towards a turbine-free region when the number of turbines surpasses a certain limit. Lastly, analyzing the effect of different wind resources it can be observed that the sites characterized by unidirectional wind do not benefit significantly from the co-design approach. In this case, the traditional methods adopted to solve the WFLOP arrange the positions of the turbines in order to avoid alignment along the dominant wind direction, limiting the necessity of applying wake steering.

The results described in these sections are limited to the conditions tested in this study, which give a broad quantitative overview of the co-design potential but leave some points of discussion that can be addressed qualitatively as follows. First, the wake models adopted in this work can influence the results of this analysis. Specifically, the adoption of a wake added turbulence model such as Crespo-Hernandez (Crespo and Hernandez, 1996) is expected to widen the wake shape and enhance its recovery. This could decrease the effectiveness of wake steering and the co-design approach in case of large wind farms. However, these choices are not expected to impact significantly neither the geometric yaw relations nor the magnitude of the AEP gains described in this study. Second, the results presented in this research refer to the chosen reference turbine, i.e. the DTU 10MW, and therefore can differ in case of a different turbine type. Nevertheless, all the geometric relations presented here are normalized with the rotor diameter, hence they are not sensitive on the size of the turbine. However, the mismatch between the geometric models, which generally scale with $D$, and the wind turbine power, proportional to $D^2$, can influence the analysis. This is expected to have a consequence on the coefficients of the geometric yaw relations, requiring additional tuning, whereas the final conclusion should not be impacted significantly.

Lastly, the analysis on the effect of the wind direction resolution adopted for AEP calculation with wake steering has highlighted the importance of this parameter for the evaluation of the co-design approach. Specifically, the simplicity of the base-case chosen to enable a straightforward comparison between the different methods has caused a high sensitivity with respect to the wind direction bin size. In general, the use of a fine resolution seems to decrease the benefits of the co-design method, increasing the uncertainty on the results presented in this work. However, whereas the magnitude of the AEP improvement is seriously affected by the size of the wind direction bins, the trends with the variables of interest are expected to remain valid. Overall, this analysis has emphasized the necessity of including uncertainties within the yaw optimization process, as suggested by the work of Quick et al. (2020). This would enable a more realistic evaluation of the co-design approach.

## 8 Conclusions

A genetic algorithm tailored to the layout optimization and referred to as LO-GA has been developed in this study, where the crossover and the mutation phases are implemented to capture the physical meaning of the optimization variables, in order to improve the exploration of the design space. This method enables the improvement of regular layouts, usually not achievable with basic versions of GA. Moreover, three novel relations have been introduced to calculate the geometric yaw angles, namely linear, exponential and exponential corrected approach. Whereas the former two methods are based only on the streamwise and cross-stream distances of the nearest turbine, the last approach consider all the downstream turbines within the wake, enabling a more accurate prediction.

A base-case consisting in a 16-turbines farms located at the HKN site has been used to calculate the improvement achieved with the co-design method, for which an increase in the AEP between $0.3\%$ and $0.4\%$ has been obtained. To evaluate the potential of the co-design approach based on wake steering, a wider range of cases have been tested and the conclusions are summarized in the following list:

– The wind farms characterized by a high power density benefit the most from the co-design approach, irrespective of the shape of the available surface where the turbines can be positioned.

– Increasing the number of turbines, an AEP increase up to $0.6\%$ can be obtained. However, this value stabilizes for a number of turbines higher than $25$.

– Sites characterized by a unidirectional wind do not benefit significantly from the co-design approach.

Besides the advantages in terms of AEP increase that we have mentioned, this study has investigated the effect of not applying the control strategy during the operation phase in a layout optimized using the co-design method. Specifically, it has been shown that a decrease in the AEP would occur, recommending that firm decisions about the control strategy have to be taken prior to the design phase. To minimize the risk of losses related to future decisions on the control strategy, a multi-objective co-design method has been proposed, for which the layout is optimized simultaneously for the case when wake

steering is applied and when this does not occur during the operation phase. Adopting this approach, the AEP losses in a 16-turbines layout can be limited to $0.1\%$ if wake steering is not adopted, while keeping the AEP gain above $0.3\%$ in case wake steering is applied.

An analysis on the effect of the wind direction resolution has shown that the magnitude of the AEP gains is significantly affected by this parameter. Decreasing the size of the wind direction bins has resulted in a negative effect on the benefit

of the co-design approach. However, such influence is partially caused by the unrealistic assumption of minimal error in wind direction measurements and absence of constraints in yaw actuation. Therefore, this analysis has raised the necessity of integrating uncertainties within the yaw optimization to provide an accurate evaluation of the co-design method, indicating an interesting pathway for future research.

Lastly, some other recommendations for future work are mentioned. First, the geometric yaw relations can be improved to

620 provide a more accurate approximation of the optimal yaw angles. Specifically, the relations developed in this study neglect the scenario when wake steering is not applied when two aligned turbines are too close to each other. Second, the integration of grey/black-box machine learning models in the co-design framework is recommended to understand if a further increase in the AEP can be achieved in case of a better approximation of optimal yaw angles. This study has shown that better predictions of the optimal yaw angles do not lead to a significant improvement in the co-design approach. However, there is still a gap

between the yaw angles obtained with the serial-refine method and the geometric yaw relations, which could be filled using models based on machine learning techniques. Third, another recommendation for future work concerns the objective of the WFLOP, which could be extended further than the AEP, for instance to load mitigation.

*Author contributions.* Matteo Baricchio: conceptualisation, methodology, software, validation, investigation, writing – original draft, visualisation. Pieter M.O. Gebraad: writing – review & editing, supervision. Jan-Willem van Wingerden: writing – review & editing, conceptualisation, supervision, resources, funding acquisition.

*Competing interests.* At least one of the (co-)authors is an Associate editor of Wind Energy Science.

*Acknowledgements.* This work has been supported by the SUDOCO project, which receives the funding from the European Union's Horizon Europe Programme under the grant No. 101122256.

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
