# Peer review of "Evaluating the potential of wake steering co-design for wind farm layout optimization through a tailored genetic algorithm"

_Wind Energy Science, 2024_

## Author Comment (AC1)

**Response to Reviewer 1**

Matteo Baricchio, Pieter M.O. Gebraad and Jan-Willem van Wingerden

Before responding, we would like to thank you for the time spent on reading our work and we highly appreciate your relevant comments that have helped us to improve the quality of the manuscript. We have included below our response to your comments, which are shown in blue. The main modifications made in the paper are included in this document in magenta.

The paper describes a new way to tackle a combined wind farm layout and wake steering optimization problem by applying a genetic algorithm. With some clever methods to handle mutation and crossover, it is able to produce better results than with a basic implementation available in PyGAD, which has trouble converging. Moreover, the paper builds on the work of Stanley et al. (2023) to avoid a nested optimization and does so by proposing a new formulation for the geometric yaw relation. The AEP gains are in the order of 0.4%, which is significant for a commercial wind farm and worth pursuing. The authors explain the method very thoroughly and included a couple of helpful drawings to help the reader understand at key points in the story. It is also very valuable that the results are presented in a statistical fashion to show the random nature of the algorithms.

Nevertheless, I would like to make some suggestions to improve the paper. First, it would be good to explain (qualitatively) how the geometric yaw expressions depend on the assumed wake shape and possibly the wind turbine type. This could explain why the expression from Stanley et al. (2023) does not translate well to this study. The tuning parameter $\gamma_{max}$ could also benefit from some introduction, especially since it can be linked to physical actuation limits and may be a first step towards a multi-objective optimization with structural loads.

Thanks for these questions and remarks. These are relevant points that have been included in the paper as follows. First, in the Discussion section a paragraph has been added to comment on the limitations of our study concerning the wake models and reference turbine chosen for our simulation. Specifically, the effect of adopting a wake added turbulence or changing the size of the reference turbine is discussed qualitatively. Second, a paragraph is added in section 4.3 to highlight the "meaning' of the $\gamma_{max}$ parameter, mentioning the possibility of adapting its value in case structural loading and/or actuation limits are taken into account.

Added paragraph in the Discussion: The results described in these sections are limited to the conditions tested in this study, which give a broad quantitative overview of the co-design potential but leave some points of discussion that can be addressed qualitatively as follows. First, the wake models adopted in this work can influence the results of this analysis. Specifically, the adoption of a wake added turbulence model such as Crespo-Hernandez (Crespo and Hernandez, 1996) is expected to widen the wake shape and enhance its recovery. This could decrease the effectiveness of wake steering and the co-design approach in case of large wind farms. However, these choices are not expected to impact significantly neither the geometric yaw relations nor the magnitude of the AEP gains described in this study. Second, the results presented in this research refer to the chosen reference turbine, i.e. the DTU 10MW, and therefore can differ in case of a different turbine type. Nevertheless, all the geometric relations presented here are normalized with the rotor diameter, hence they are not sensitive on the size of the turbine. However, the mismatch between the geometric models, which generally scale with $D$, and the wind turbine power, proportional to $D^2$, can influence the analysis. This is expected to have a consequence on the coefficients of the geometric yaw relations, requiring additional tuning, whereas the final conclusion should not be impacted significantly.

Added paragraph in the section 4.3: Among the various coefficients present in the geometric yaw rela-

[Figure]

Figure 1: Regular layout for a 16-turbines wind farm

tions and included in Table 4, $\gamma_{max}$ has a clear physical interpretation since it represents the maximum absolute value for the yaw angles of the turbines. In this study, $\gamma_{max}$ has been tuned targeting the maximization of the AEP without applying any restriction. However, the tuning of $\gamma_{max}$ could be constrained to take into account actuation limits or requirements on the structural loading, since it has been demonstrated that wake steering can have a negative impact on some load channels (Shaler et al., 2022).

Furthermore, with 36 wind direction bins (i.e., 0, 10, ..., 350°), there may be a danger that the optimizer is gaming the cost function by orienting the turbines along the edges of the wind direction bins (i.e., along 5, 15, ..., 355°). Since the AEP increases are only $< 0.5\%$, this could play a role. Have the authors checked the resulting layouts for such artefacts and whether they impact the result significantly?

Thank you so much. We did not realize this aspect and we did an extra study concerning the sensitivity on the wind direction resolution, i.e. the size of the wind direction bins. This turned out to provide significant insights related to our work. Therefore, we have added a new section (6.6) in the manuscript where this analysis is described. In the following paragraphs we will provide further comments to motivate and clarify the content added to the manuscript.

In summary, the base-case described in section 5.2 has been simulated with different wind direction resolutions, affecting both the layout-optimization phase and the subsequent yaw optimization performed through the serial-refine method. Decreasing the size of the wind direction bins, i.e. using a finer resolution for the wind direction, a drop in the AEP improvement has been obtained, as shown in the added Fig. 19 of the manuscript.

To gain further insights on this behaviour, a toy example consisting of 16-turbines in a regular 4x4 layout is studied, keeping the other characteristics of the base-case (e.g. power density 20 W/m²). This example is chosen since the directions of the main wake losses are clear and intuitive. Therefore, this layout is shown in Fig. 1 whereas Fig. 2 indicates the increase in the power generation (at a wind speed of 8 m/s) when wake steering is applied for different wind directions (x-axis) and bin sizes used for the discretization (different colors).

As we have mentioned in section 6.6, we have identified two main effects in this trend.

**Drop in AEP from 10 deg to 5 deg.** This is caused by the limitations of our base-case, whose simplicity enhances the influence of this parameter. Specifically, the wind direction of 45 deg is not simulated when a bin size of 10 deg is used. Since for both the co-design and the sequential layouts the turbines tend to be positioned at the borders of the domain to maximize the area usage, the directions that are multiples of 45 deg are still the ones characterized by the highest wake losses. Therefore, when the 5 deg bin size is used, such directions are all included in the AEP calculation, increasing the total

[Figure]

Figure 2: Power increase obtained by applying wake steering for different wind directions and bin sizes. A wind speed of 8 m/s is considered.

wake losses of the farm and therefore decreasing the AEP values. Even though this effect regards both the sequential and the co-design layouts, it results in an overall reduction of the AEP improvement. As mentioned in the discussion and the conclusion of the manuscript, we believe that this effect can increase the uncertainty concerning the magnitude of the AEP gains mentioned in the paper. However, the trends between the different geometric yaw relations and the relevant variables (e.g. power density, number of turbines, wind direction variability) are expected to remain valid. Moreover, this effect is expected to have minor impact in case larger layouts are simulated, where the wake losses would be more uniform along the wind directions.

**Drop in AEP improvement from 5 deg to lower bin sizes.** In this case another behaviour is observed: whereas the AEP of the co-design layouts seems not to be affected by the wind direction bin size, in case of the sequential layouts the AEP values increase as the wind direction resolution gets finer. As a consequence, the AEP improvement of the co-design approach tend to converge to 0% as the bin size decreases. In the example that we have provided about the regular layout (Fig. 1 and 2), it can be observed that the main power increase caused by wake steering occurs in the neighbourhood of the directions characterized by the highest wake losses (not in that exact direction), therefore simulating such directions leads to a relevant increase in the AEP. For this reason, using smaller wind direction bins while performing the yaw optimization (serial-refine method) causes an increase in total AEP value. The substantial magnitude of this effect can be detected in Fig. 2, especially for the wind direction of 90 deg. Therefore, this effect becomes dominant with respect to the benefits of co-design approach, which would be ineffective if such conditions were applied. However, as we have mentioned in the paper, these conditions require very precise wind direction measurements and minimal yaw offset errors, which are far from the real possibilities. In conclusion, this analysis enhances the necessity of adopting a yaw optimization method that includes the uncertainty on the wind direction if we want to understand the actual potential of the co-design approach, indicating the pathway for a future research.

We believe that this conclusion is now a fundamental part of our work and we would like to thank you again for your comments and remarks. Therefore, we have mentioned this aspect also in the Abstract, the Discussion and the Conclusions of the manuscript.

Added section:

[revised manuscript text omitted]

The discussion section would benefit from some reflection on the 15-W/m² saturation point. Particularly whether it is a universal limit or case-specific – would one expect the same with different wind farm plot shapes, or is this simply driven by the turbine spacing and how much a wake can actually be deflected from the downwind turbine?

We agree that some additional analysis about the presence of this saturation point can add significant value to our work. Therefore, the sensitivity analysis on the power density has been repeated for two additional shapes of the available surface. These additional results have been included and discussed in section 6.4.1. As mentioned in the manuscript, this saturation point at 15 W/m² is indeed dependent on the domain shape. Therefore, slight modifications have been made to the Abstract, Discussion and Conclusions of the manuscript. In these sections, it is emphasized that the decrease in the AEP improvement for low power densities persists regardless of the domain shape, rather than indicating a precise value as saturation point.

Added paragraph in section 6.4.1: Further analysis is performed to investigate whether the saturation limit of $15\,\mathrm{Wm}^{-2}$ is dependent on the squared shape of the domain or it remains valid for other geometries. For this purpose, the simulations are repeated for two different rectangular areas, characterized by a ratio between the sides equal to 1.5 and 2.0, respectively. The results are limited to the exponential corrected relation and are included in Fig. 4. It can be observed that the magnitude of the AEP gains decreases and the saturation behavior identified in the previous case is not evident anymore, concluding that such results are dependent on the shape of the area where the turbines can be positioned. However, the decreasing trend for the AEP gains in case of lower power density values remains valid, hence it can be considered a general conclusion for this sensitivity analysis.

[Figure]

Figure 4: AEP gain obtained with co-design method (exponential corrected relation) for different power densities, showing the trend for different shapes of the available surface, identified through the ratio between the sides of a rectangle. The areas surrounding the lines (median values) refer to the range between the 25th and 75th percentiles, being the results of multiple simulations of the same case.

Some other minor points that may be considered for a next revision:

- There are a couple mix-ups between US vs. British English (e.g., recognize vs. recognise, favorable vs. favourable). Thanks for pointing it out. It has been fixed in the manuscript.

- For one who wants to reproduce the work, it would be helpful to add a reference to the HKN site conditions shown in Fig. 8 (they are public). We agree. A reference about the wind resource data of HKN has been added in section 5.1.

- The shaded areas in Fig. 10 are very light and difficult to read from screen. Thanks for the comment. The color of the shaded areas has been made darker both in Fig. 1 and Fig. 10.

- It is very difficult for a reader who quickly browses through the paper to understand what is meant with "Std of the probability of occurrence" (Fig. 16). I would suggest renaming the axis label to "wind direction variability" and/or spent a line in the caption to explain. We agree. The x-axis of the figure has been changed according to your suggestion and additional explanation has been included in the caption.

- It would be helpful to state what happens with the area of the wind farm plot under different power densities and higher numbers of turbines in Section 6.4.1 and 6.4.2, respectively. Now it is not immediately clear whether the wind farm plot increases in size with an increasing number of turbines. We agree on this point and to make this step clear some additional explanations have been included in section 5.3.

- I feel that renaming the sections 6.4.2 and 6.4.3 to simply "Number of turbines" and "Wind direction variability" woud reflect the contents better. A browsing reader could also connect those better to the axis labels of Fig. 15 and 16. Thanks for the suggestion. The titles of these sections have been changed accordingly.

Overall, nice work and I hope to see the final paper soon!

Thanks! We really appreciate your feedback.

---

## Author Comment (AC2)

**Response to Reviewer 2**

Matteo Baricchio, Pieter M.O. Gebraad and Jan-Willem van Wingerden

Before we proceed with our reply, we want to express our gratitude for reading our work and contributing to further improve the content of our paper and future work. We have included below our response to your comments, which are shown in blue. The main modifications made in the paper are included in this document in magenta.

Thank you for this contribution! In general, writing is well done and figures are clear. References to literature is thorough. The innovation of optimizing for the coupled condition that wake steering is or isn't applied to the co-designed farm is a valuable contribution.

Thanks for the feedback! We truly appreciate it.

Figure 17: I understand the pareto front for the multi-objective points, and for the co-design points, I understand they improve with wake steering, but lead to losses without. The sequential points are more mysterious to me, there are some that appear to lose AEP with and without wake steering applied?

Thanks for your comment, we agree that additional explanation is required. Multiple points are included in Fig. 18 (Fig. 17 in the previous version of the manuscript) also for the sequential and the co-design methods as a result of multiple simulations of the same case. This strategy is adopted to prevent a bias in the results caused by the random nature of the genetic algorithm. Therefore, whereas in practice only the best solution is chosen, in our work multiple solutions are presented to enable a probabilistic interpretation of the results. In particular, in this figure there are some solutions for the sequential approach (green dots) that are clearly better than the other solutions obtained with this approach for both objectives, however, all of them are included for the reason explained in this paragraph. To clarify this concept to the reader, an additional explanation is added in the caption of the figure, mentioning that multiple solutions are included also for the sequential and co-design approaches to keep the consistency with the other plots.

Modified caption of Fig. 18: Multi-objective co-design approach based on the multi-objective optimization. Each data point indicates an optimized layout. Multiple data points are included also for the sequential and the co-design approach, resulting from different simulations and providing a probabilistic interpretation of the results in accordance with the other graphs. The axis of the plot refer to the AEP increase with respect to the average AEP value obtained through the traditional sequential approach. The Pareto front is present on the top-right of the plot, resulting from a max-max problem. A cross is included to identify a possible robust solution.

The algorithm for the geometric yaw of Stanley is also implemented and publicly availble within FLORIS: (see for example here: `https://github.com/NREL/floris/blob/main/floris/optimization/yaw_optimization/yaw_optimizer_geometric.py` and here: `https://nrel.github.io/floris/examples/examples_control_optimization/006_compare_yaw_optimizers.html`)

Thanks for pointing this out. The availability of this algorithm in FLORIS has been mentioned in the manuscript, adding this information to the Introduction as follows: This approximation of the optimal yaw angles has also been implemented in the open-source software FLORIS (National Renewable Energy Laboratory, 2024).

This could be interesting to point out. Figure 15 also gives the impression that these improvements

can be fairly critical, and perhaps one of the improved algorithms from this paper could be submitted as a pull request back to FLORIS.

Thanks for your suggestion. We are working on decoupling these algorithm from the software that we have adopted in this work, i.e. PyWake, and we will be keen to integrate them into FLORIS. Therefore, we will soon submit a pull request to FLORIS for this purpose.